# Structural insight into ligand binding and activation of the orphan GPCR Mas1

Yumu Zhang [1,2,5✉], Qiuying Wang [3,4,5], Heng Liu [1,2,5], Hong Shan [1,2], Yimin Gu [1,2], Jiaqi Yang [1,2], Yuan Gao [1,2], Kai Wu [1], Dehua Yang [2,4✉] & H Eric Xu [1,2,3✉]

## Abstract

The Mas1 receptor, an orphan class A G-protein-coupled receptor (GPCR), plays pivotal roles in cardiovascular and anti-inflammatory regulation. Despite its therapeutic relevance, the structural mechanisms underlying Mas1 ligand binding and activation remain poorly understood. Here, we report cryo-EM structures of Mas1 bound to two chemically distinct agonists—neuropeptide FF (NPFF) and synthetic small-molecule AR234958—captured in complex with inhibitory G proteins. These structures reveal a conserved orthosteric binding pocket accommodating both ligands through shared hydrophobic interactions. Unlike many other class A GPCRs that rely on direct $W^{6.48}$ toggle switch engagement, Mas1 adopts a non-canonical activation strategy driven by a ligand-induced hydrophobic compression plane involving residues $Y248^{6.55}$, $L87^{2.60}$, $I84^{2.57}$, and $L266^{7.39}$ at the bottom of the ligand binding pocket. This mechanism transmits mechanical tension to promote TM6 displacement and G protein coupling. Functional mutagenesis validates this model, identifying two transmembrane helix 6 (TM6) residues, $M244^{6.51}$ and $F237^{6.44}$, as critical molecular switches. Comparative analyses of Mas1-related receptors, MRGPRX1–X4, reveal conserved features and mechanistic divergence within this subfamily. These findings provide a structural framework for understanding Mas1 pharmacology and rational design of selective therapeutics.

**Keywords** GPCR; NPFF; Mas1; AR234958; Cryo-EM Structures
**Subject Categories** Signal Transduction; Structural Biology

## Introduction

The Mas1 proto-oncogene encodes a G protein-coupled receptor (GPCR) that plays a pivotal role in the alternative axis of the renin–angiotensin system (RAS) (Kostenis et al, 2005; Santos et al, 2003). Unlike the classical angiotensin II type 1 receptor (AT1R),

which mediates vasoconstriction, inflammation, and fibrosis, Mas1 primarily interacts with angiotensin-(1–7) [Ang-(1–7)], a heptapeptide exerting vasodilatory, anti-inflammatory, and antifibrotic activities (Santos et al, 2018; Simões e Silva et al, 2013). This ACE2/Ang-(1–7)/Mas1 axis acts as a key counter-regulatory pathway, which plays a crucial role in maintaining cardiovascular, renal, and metabolic homeostasis (Tikellis and Thomas, 2012).

As a class A orphan GPCR, Mas1 exhibits distinctive pharmacological properties and signaling mechanisms. Specifically, Mas1 can be activated by other ligands, such as neuropeptide FF (NPFF), which modulates pain perception, cardiovascular function, and neuroendocrine regulation (Nguyen et al, 2020; Yang et al, 1985). Additionally, the small-molecule agonist AR234958 has been identified as a selective pharmacological activator of Mas1, further broadening the therapeutic potential of this receptor (Gaidarov et al, 2018).

Mas1 activation triggers complex downstream signaling networks through multiple G protein pathways (Kostenis et al, 2005). The receptor predominantly couples to inhibitory G-proteins (Gαi/o), resulting in reduced intracellular cAMP and suppression of proinflammatory cascades (Burghi et al, 2017; Sampaio et al, 2007). However, multiple evidence indicates that Mas1 can also couple to Gαq/11 and Gα12/13 proteins in specific cellular contexts, triggering intracellular calcium flux, RhoA activation, and cytoskeletal reorganization (Chikumi et al, 2002; Gavard and Gutkind, 2008; Sureshkumar et al, 2023). These diverse coupling mechanisms ultimately lead to increased nitric oxide (NO) production, PI3K/Akt pathway activation, and MAPK/NF-κB signaling suppression, collectively mitigating oxidative stress and inflammatory responses (Santos et al, 2018; Yan et al, 2024).

The physiological importance of Mas1 signaling is underscored by accumulating evidence of its protective roles in diverse pathological conditions, including hypertension, myocardial infarction, diabetic nephropathy, and neurodegenerative disorders (Brito-Toscano et al, 2023; Molaei et al, 2023; Santos et al, 2018; Xie et al, 2022; Xue et al, 2021; Zhou et al, 2022).

Despite this growing recognition of Mas1's therapeutic importance, the receptor remains structurally understudied compared to other RAS components. Critical gaps persist in our understanding of its molecular architecture, ligand selectivity mechanisms, and

[1] State Key Laboratory of Drug Research, Center for Structure and Function of Drug Targets, Shanghai Institute of Materia Medica, Chinese Academy of Sciences, Shanghai, China. [2] University of Chinese Academy of Sciences, Beijing, China. [3] School of Chinese Materia Medica, Nanjing University of Chinese Medicine, Nanjing, China. [4] State Key Laboratory of Chemical Biology and The National Center for Drug Screening, Shanghai Institute of Materia Medica, Chinese Academy of Sciences, Shanghai, China. [5] These authors contributed equally: Yumu Zhang, Qiuying Wang, Heng Liu. ✉E-mail: yumu_zhang@hms.harvard.edu; dhyang@simm.ac.cn; eric.xu@simm.ac.cn

detailed activation pathways. Recent advances in GPCR structural biology now provide unprecedented opportunities to elucidate these fundamental mechanisms and guide rational drug design efforts targeting the protective ACE2/Ang-(1–7)/Mas1 axis (Tamargo and Tamargo, 2017).

Here, we address this knowledge gap by reporting the first cryo-electron microscopy (cryo-EM) structures of Mas1 bound to two chemically distinct agonists—neuropeptide FF (NPFF) and the synthetic small-molecule AR234958—each captured in complex with Gαi proteins. Our structural analyses reveal the molecular basis of Mas1's ligand promiscuity and uncover a non-canonical activation mechanism that distinguishes this receptor from classical GPCRs. These findings establish a comprehensive structural framework for understanding Mas1 pharmacology and provide new opportunities for developing selective therapeutics targeting this important but understudied receptor system.

# Results

## Ligand recognition and activation of Mas1

To investigate the structural basis of Mas1's ability to recognize chemically diverse ligands, we first examined the relationship between Mas1 and related receptors within the MRGPR subfamily. Sequence alignment analysis reveals that Mas1 clusters with MRGPRX1-X4 receptors (Dong et al, 2001), forming a distinct branch of class A GPCRs (Appendix Fig. S1). This evolutionary relationship suggests shared structural features that may underlie common ligand recognition mechanisms within this receptor family.

Despite the chemical diversity of Mas1 ligands—ranging from the heptapeptide Ang-(1–7) to the Octapeptide NPFF and the synthetic small molecule AR234958—sequence alignment analysis revealed striking structural commonalities among the peptide ligands. Most notably, both NPFF and Ang-(1–7) contain conserved aromatic-rich motifs that appear critical for receptor recognition (Fig. 1A). NPFF exhibits a characteristic "F-LF" motif (phenylalanine-leucine- phenylalanine) at positions 1–3, while Ang-(1–7) contains a similar aromatic signature with key valine and tyrosine residues. This conservation of aromatic character suggests that hydrophobic and π–π stacking interactions may be fundamental to peptide recognition by Mas1.

To validate these structural predictions and assess the functional consequences of ligand binding, we performed comprehensive dose–response analyses using cAMP accumulation assays. All three ligands—NPFF, Ang-(1–7), and AR234958—effectively activated Mas1 in a concentration-dependent manner, confirming their agonistic properties (Fig. 1B). However, significant differences in potency were observed, with AR234958 demonstrating superior efficacy compared to the peptide ligands. Specifically, AR234958 achieved half-maximal activation at approximately 6.71 ($pEC_{50}$), while NPFF and Ang-(1–7) required higher concentrations for equivalent responses, with $pEC_{50}$ values of 4.98 and 3.13, respectively (Fig. 1B; Appendix Table S1). Though these potency differences suggest that while all ligands engage the same receptor through shared aromatic interactions, structural differences in binding mode or receptor stabilization may account for their distinct pharmacological profiles.

## Cryo-EM analysis and overall structure

To elucidate the molecular determinants of ligand recognition and receptor activation in Mas1, we determined high-resolution cryo-electron microscopy (cryo-EM) structures of the Mas1 receptor in complex with two chemically distinct agonists: the neuropeptide FF (NPFF) and the synthetic small-molecule AR234958, each bound to heterotrimeric Gαi proteins (Fig. 2A,B). To enable efficient expression and purification, we engineered the wild-type full-length Mas1 receptor with an N-terminal hemagglutinin (HA) epitope tag, a 6×His tag for affinity purification, and a bRIL fusion tag. Co-expression of Mas1 with Gαi heterotrimers and the stabilizing antibody fragment scFv16, followed by incubation with NPFF or AR234958, facilitated the assembly of homogeneous receptor–ligand–G protein complexes suitable for structural analysis.

The cryo-EM structures of the NPFF–Mas1–Gαi–scFv16 and AR234958–Mas1–Gαi–scFv16 complexes were resolved at global resolutions of 2.54 Å and 3.07 Å, respectively (Appendix Figs. S2 and S3). These high-resolution maps provided clear density for most residues within the ligands, receptor (Appendix Fig. S4), and G protein components, enabling detailed visualization of the receptor–ligand interfaces and G protein interactions. Structural analysis revealed that Mas1 adopts a canonical seven-transmembrane (7TM) GPCR architecture, consistent across both ligand-bound states (Fig. 2A,B). The minimal conformational differences between the NPFF- and AR234958-bound complexes suggest a conserved mechanism of Gαi coupling, despite the chemical diversity of the ligands, highlighting Mas1's structural adaptability in mediating downstream signaling.

A defining feature of Mas1's ligand recognition is its shallow, hydrophobic orthosteric binding pocket, formed by residues from transmembrane helices TM2, TM3, TM5, TM6, and TM7, as well as extracellular loop 2 (ECL2) (Fig. 2C). This cavity is lined by an extensive network of hydrophobic and aromatic residues, including P33[1.33], H36[1.36], I84[2.57], Y91[2.64], I105[3.25], Y102[3.22], F112[3.32], L113[3.33], Y168[4.61], I172[ECL2], Y248[6.55], Y251[6.58], Y252[6.59], H262[7.35], H263[7.36], and L266[7.39] (Ballesteros–Weinstein nomenclature (Ballesteros and Weinstein, 1995)) (Fig. 2C). These residues create a predominantly non-polar environment that stabilizes both peptide and small-molecule ligands. This conserved pocket architecture underpins Mas1's ability to accommodate chemically distinct agonists, providing a structural foundation for its ligand promiscuity and activation, as further explored in subsequent sections.

## Binding modes of NPFF for Mas1

The cryo-EM structure of the Mas1 receptor in complex with neuropeptide FF (NPFF) at 2.54 Å resolution reveals a detailed molecular framework for ligand recognition within a hydrophobic orthosteric pocket formed by transmembrane helices TM1–TM3, TM4–TM7, and extracellular loop 2 (ECL2) (Fig. 3A). NPFF adopts a flexible binding conformation, driven by three key residues—phenylalanine 1 (1F) (Fig. 3B), leucine 2 (2L) (Fig. 3C), and phenylalanine 3 (3F) (Fig. 3D)—which engage distinct sub-pockets within the receptor's transmembrane cavity, facilitating stable ligand anchoring and receptor activation (Fig. 3A).

To dissect the roles of these residues, we designate 1F as the secondary hydrophobic core (SHC), which establishes peripheral

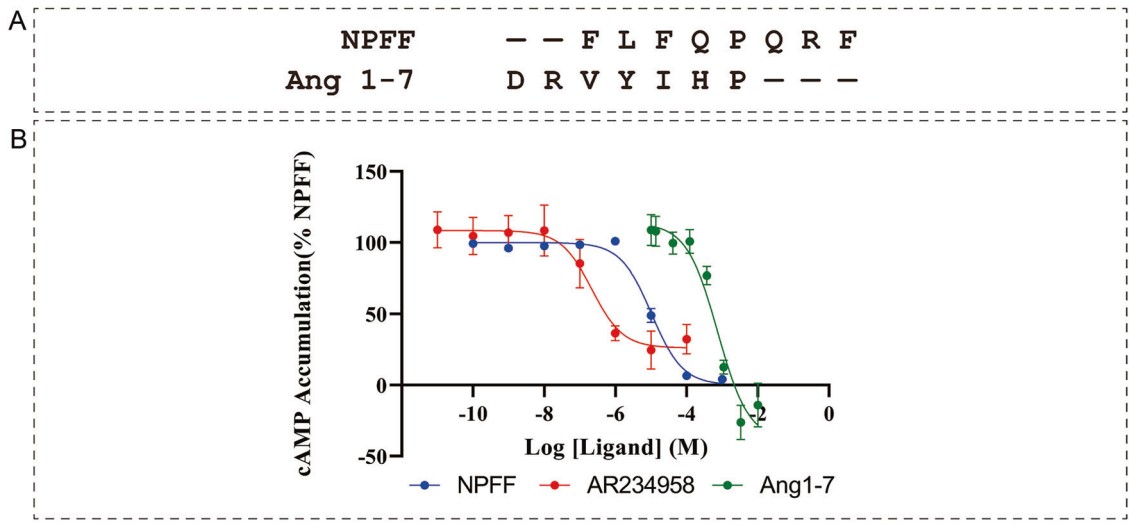

**Figure 1.  Mas1 responds to NPFF, AR234958 and shows low-potency response to Ang-(1–7) in HEK293T cells.**

(A) Primary amino acid sequences of NPFF and Ang 1–7 peptides. Conserved and functionally analogous residues are aligned to highlight structural similarities that may underlie Mas1 binding. (B) Dose–response curves of cAMP inhibition in HEK293T cells transiently expressing Mas1. Cells were stimulated with forskolin and treated with increasing concentrations of NPFF (blue), AR234958 (red), or Ang 1–7 (green). cAMP accumulation was measured and normalized to NPFF-induced maximal response. Data are presented as mean ± SEM from three independent experiments ($n = 3$). Curves were fitted using nonlinear regression. Source data are available online for this figure.

hydrophobic contacts to guide initial ligand orientation (Fig. 3A). In contrast, 2L and 3F collectively form the central hydrophobic core (CHC), serving as the primary interaction hotspot that anchors NPFF within the binding pocket (Fig. 3A). Specifically, 1F inserts into a hydrophobic sub-pocket defined by Mas1 residues I172$^{ECL2}$, Y168$^{4.61}$, while the hydroxyl group of S109$^{3.29}$ forms a critical hydrogen bond with the backbone amide nitrogen of NPFF-Phe1 to anchor the ligand orientation (Fig. 3B). Functional mutagenesis confirmed the critical role of these residues, alanine substitutions of Y168$^{4.61}$ and S109$^{3.29}$ abolished NPFF potency, whereas I172A primarily reduced maximal efficacy (Emax) with minimal change in EC50, highlighting their importance in stabilizing ligand binding and initiating receptor activation (Fig. 3E).

The central 2L residue bridges these sub-pockets, interacting closely with Y248$^{6.55}$, while H262$^{7.35}$ defines the structural boundary of the sub-pocket to support NPFF's conformation (Fig. 3C). Mutagenesis of Y248$^{6.55}$ and H262$^{7.35}$ markedly reduced ligand efficacy. Notably, although flow cytometry confirmed robust surface expression of the H262A mutant (~90% of WT, Appendix Table S2), it caused a drastic reduction in maximal response (Emax) without significantly altering ligand potency (EC50). This phenotype suggests that H262 is critical for maintaining the active pocket conformation or transduction efficiency, rather than mediating direct high-affinity ligand contacts. The 3F residue of NPFF engages a deeper hydrophobic sub-pocket comprising Mas1 residues I39$^{1.39}$, I84$^{2.57}$, L87$^{2.60}$, Y91$^{2.64}$, L266$^{7.39}$, as well as H36$^{1.36}$ and H263$^{7.36}$.(Fig. 3D). Alanine mutations at these positions (I39A, I84A, L87A, Y91A, L266A) similarly impaired receptor activation, underscoring their role in securing the ligand's aromatic core and supporting signal transduction (Fig. 3G). The H36A mutation primarily reduced maximal response (Emax) without significantly

shifting ligand potency (Fig. 3G; Appendix Table S2). Interestingly, the H263A mutation had minimal impact on receptor activity (Appendix Table S2). This suggests that while H36$^{1.36}$ structurally lines the pocket, its role—analogous to H262$^{7.35}$—is likely centered on maintaining the active pocket conformation rather than determining ligand binding affinity. Furthermore, although H263$^{7.36}$ is spatially proximal, it appears functionally dispensable for NPFF recognition.

To assess the reliability of computational modeling, we compared the cryo-EM-derived Mas1–NPFF structure with an AlphaFold3 (AF3)-predicted model of the same complex (Fig. 3H). Both structures exhibit a conserved ligand-binding pocket architecture, with NPFF occupying analogous spatial regions and key residues (e.g., Y248$^{6.55}$, L266$^{7.39}$, L87$^{2.60}$, I84$^{2.57}$) aligning closely. However, the NPFF backbone orientation diverges from the EM structure, with AF3 predicting altered side-chain positions for residues 4Q and 5P to the other side of the pocket. Despite these discrepancies, the conserved hydrophobic core motif centered around the F-L-F residues remains similar in both models, indicating that AF3 largely captures essential hydrophobic anchoring determinants, but the fine-grained peptide geometry still requires experimental validation.

To contextualize Mas1's ligand recognition within the MRGPR subfamily, we compared the NPFF–Mas1 complex with structures of MRGPRX1 bound to BAM8-22 and MRGPRX2 bound to C14 (Fig. 3I). While all three receptors share a hydrophobic binding pocket, their ligand-binding poses differ significantly. In MRGPRX1 and MRGPRX2, ligands are positioned closer to TM4 and TM5, whereas NPFF in Mas1 is oriented toward TM1 and TM2 (Fig. 3I). This spatial divergence likely contributes to distinct ligand selectivity and signaling profiles across the subfamily, reflecting specialized adaptations within their binding pocket architectures.

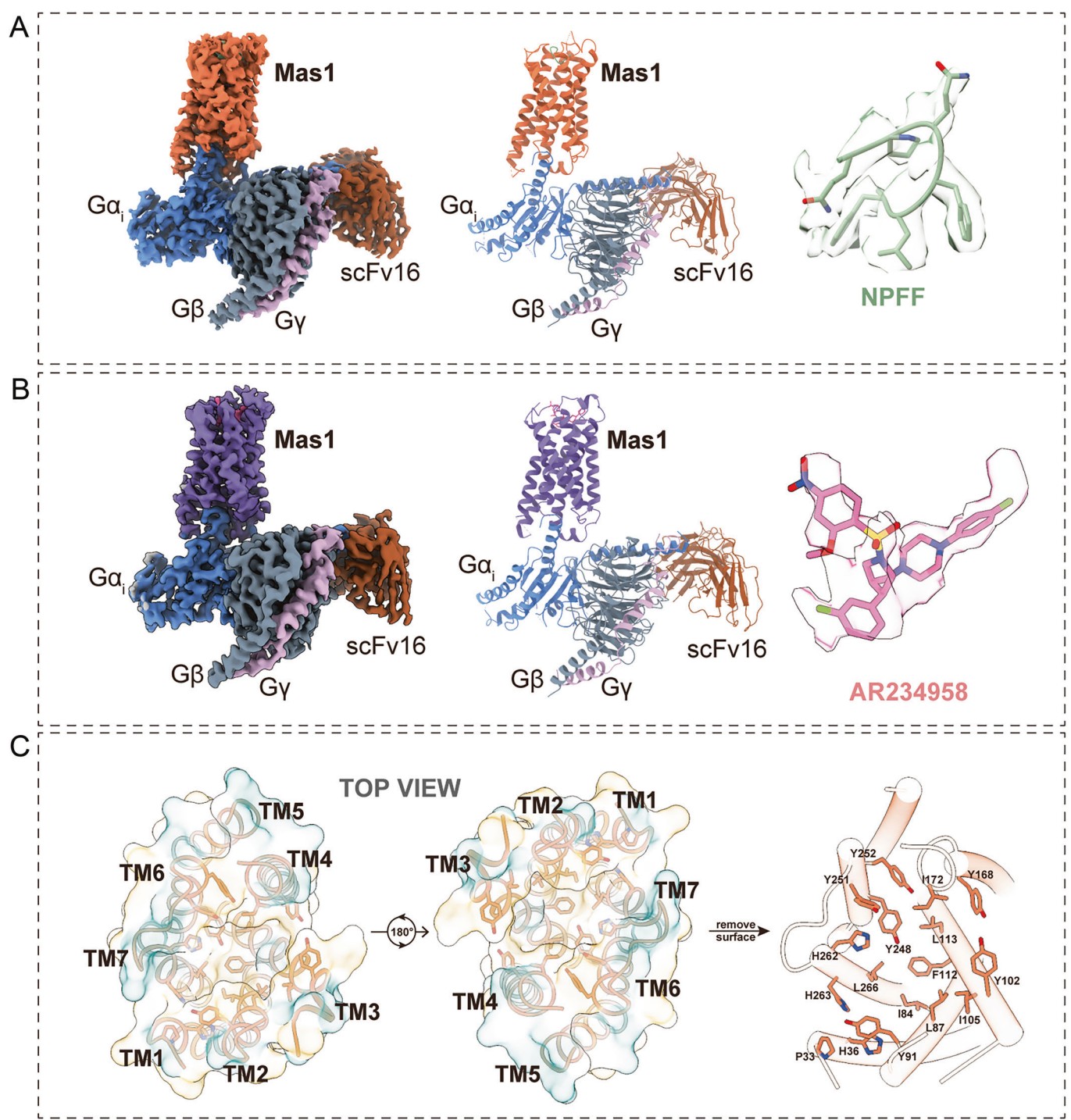

**Figure 2.    Cryo-EM structures of Mas1 reveal ligand recognition and a predominantly hydrophobic binding pocket.**

(**A**) Cryo-EM structure of the Mas1-Gi–scFv16 complex bound to NPFF at 2.54 Å resolution. Left: cryo-EM density map of the full signaling complex with individual components shown in distinct colors (Mas1: orange, Gα: gray, Gβ: dark blue, Gγ: purple, scFv16: brown). Middle: ribbon diagram of the resolved complex. Right: modeled conformation of NPFF (green), occupying the orthosteric site of Mas1. (**B**) Cryo-EM structure of the Mas1–Gi–scFv16 complex bound to AR234958 at 3.07 Å resolution. Left and middle: density map and structural model of the complex, with Mas1 shown in purple. Right: atomic structure of the synthetic ligand AR234958 (pink), indicating its accommodated pose within the orthosteric binding pocket. (**C**) Top-down views of Mas1 from the extracellular side, illustrating the ligand-binding cavity formed by transmembrane helices (TM1–TM7). Left and middle: surface representation and ribbon models at two perpendicular angles. Right: cavity-lining residues are shown in stick representation, emphasizing the hydrophobic environment of the orthosteric pocket, which is predominantly composed of non-polar side chains (e.g., F112, L87, I84, Y251, L266).

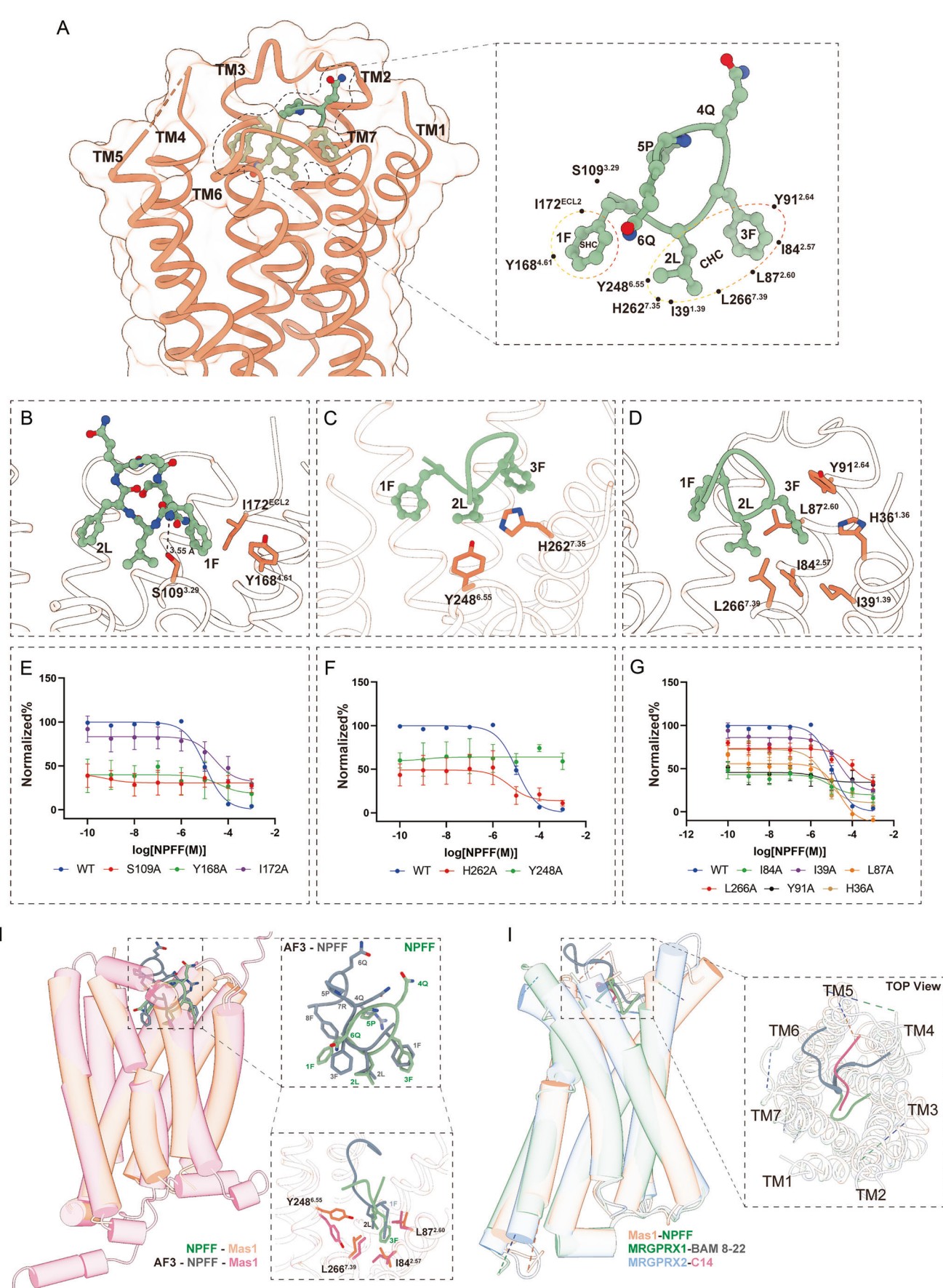

**Figure 3.    Structural and functional dissection of the NPFF binding pocket in Mas1.**

(A) Cross-sectional view of the Mas1 transmembrane (TM) domain showing NPFF (green) bound within a predominantly hydrophobic orthosteric pocket. Right: close-up of NPFF within the binding site, with key residues labeled, including I172[ECL2], Y168[4.61], S109[3.29], Y248[6.55], H262[7.35], L266[7.39], I84[2.57], L87[2.60], I39[1.39], H36[1.36] and Y91[2.64]. Ligand–receptor interactions are mediated largely through hydrophobic contacts, complemented by polar anchoring residues. (b–d) Structural snapshots highlighting specific ligand-interacting residues: (B) Contacts at the extracellular entrance (e.g., the hydrogen bond between S109[3.29] and the backbone nitrogen of NPFF-1F) (e.g., S109[3.29], I172[ECL2]). (C) Mid-pocket interactions along TM5–TM6 (e.g., Y248[6.55], H262[7.35]). (D) Basal pocket residues contributing to ligand stabilization (e.g., L266[7.39], Y91[2.64], L87[2.60], I84[2.57], I39[1.39], H36[1.36]). (E–G) Functional characterization of individual residues via cAMP assays in HEK293T cells expressing wild-type (WT) or Mas1 mutants. (E) Mutation of S109A, Y168A, or I172A abolishes ligand response. Data are presented as mean ± SEM from three independent experiments (n = 3). (F) Disruption of Y248[6.55] or H262[7.35] reduces NPFF efficacy. Data are presented as mean ± SEM from three independent experiments (n = 3). (G) Alanine substitutions of hydrophobic residues (L266A, L87A, I84A, Y91A, I39A) significantly attenuate NPFF-mediated Gi signaling. Data are presented as mean ± SEM from three independent experiments (n = 3). (H) Structural overlay of cryo-EM–resolved NPFF-bound Mas1 (salmon) and AlphaFold3-predicted NPFF–Mas1 model (gray). Insets show diverging conformations of NPFF and surrounding residues (e.g., Y248[6.55], L266[7.39], L87[2.60]), indicating that predicted models may not fully capture ligand-induced conformational states. (I) Comparative analysis of ligand positioning across Mas1–NPFF (salmon), MRGPRX1–BAM-B22 (green), and MRGPRX2–C14 (blue). Right: top view reveals distinct spatial distributions of bound ligands within the transmembrane domains, suggesting divergent ligand-recognition strategies among MRGPR family receptors. Source data are available online for this figure.

## Binding modes of AR234958 for Mas1

To explore the structural basis of Mas1's ability to recognize diverse ligands, we resolved the cryo-EM structure of Mas1 in complex with the synthetic small-molecule agonist AR234958 at 3.07 Å resolution (Fig. 4A; Appendix Fig. S3). AR234958 occupies a shallow, hydrophobic orthosteric binding pocket formed by residues from transmembrane helices TM2, TM3, TM6, TM7, and extracellular loop 1 (ECL1) (Fig. 4A). Structural analysis revealed that AR234958 comprises three key aromatic groups—Fluorophenyl Arm-1 (FA1), Fluorophenyl Arm-2 (FA2), and Surface-Accessible Phenyl Group-1 (SA1)—along with two folding cores, the 6'Benzo-folding core (6'Fc) and the 5'Purin-folding core (5'Fc), each engaging distinct hydrophobic sub-pockets within the receptor (Fig. 4A).

AR234958 engages the Mas1 orthosteric pocket through a network of hydrophobic and aromatic contacts spanning TM2, TM6, TM7 and ECL1 (Fig. 4B–D). The FA2 moiety is cradled at the extracellular mouth by I172[ECL2] and Y168[4.61], Although the refined model indicates a distance of ~5 Å between FA2 and Y168[4.61], alanine substitutions of these residues (Y168A, I172A) abolished ligand-induced activation (Fig. 4E). This demonstrates that Y168[4.61] and I172[ECL2] are essential for maintaining the stability of the pocket boundary, thereby facilitating correct ligand positioning within the core (Fig. 4B). The FA1 moiety inserts the pocket bottom, packing against I84[2.57], L87[2.60], L266[7.39] and Y248[6.55] (Fig. 4C,D). On the opposite side, SA1 moiety forms a hydrophobic network with Y91[2.64] and H262[7.35] (Fig. 4C). Specifically, one of the oxygen atoms of the sulfonyl group on SA1 acts as a hydrogen bond acceptor, forming a robust hydrogen bond with the hydroxyl group of the Y95[ECL1] side chain.

To further validate the binding stability and rationalize the local electron density quality, we performed triplicate independent 200-ns molecular dynamics (MD) simulations of the Mas1–AR234958 complex. The simulations demonstrated that the ligand maintains a stable binding pose within the orthosteric pocket (Appendix Fig. S5a). Notably, structural mapping of the ligand flexibility onto the cryo-EM density map (Appendix Fig. S5b) revealed that the regions of high mobility correspond precisely to the areas of weaker electron density. This observation is further corroborated by per-atom Root Mean Square Fluctuation (RMSF) analysis, which identified significant conformational flexibility in the peripheral aromatic moieties (Appendix Fig. S5c). Given this dynamic

behavior, the structural model in these flexible regions should be interpreted with the acknowledged limitation of local conformational heterogeneity.

Functional mutagenesis validated these structural observations. Alanine substitutions at I172[ECL2] or Y168[4.61] abolished AR234958-induced receptor activation in cAMP inhibition assays, confirming their essential role in regulating ligand access to the binding pocket (Fig. 4E). Mutations of H262[7.35], Y248[6.55] or L266[7.39] significantly reduced receptor responsiveness (Fig. 4F). Similarly, despite comparable expression levels to the wild-type (Appendix Table S3), the H262A mutant exhibited significantly impaired efficacy (Emax) for AR234958. This consistent functional impairment across chemically distinct ligands further validates H262[7.35] as a key structural determinant for the general activation mechanism of Mas1. Alanine substitutions at I84[2.57], L87[2.60], and Y91[2.64] abolished AR234958 potency, whereas Y95A primarily reduced maximal response with little effect on EC50, underscoring their supportive roles in stabilizing peripheral ligand interactions (Fig. 4G). These findings elucidate the molecular determinants of AR234958 recognition, revealing a conserved hydrophobic framework that complements the NPFF-binding mechanism and supports Mas1's ligand promiscuity, as explored further in comparative analyses (Fig. 5).

## Comparative structural analysis of NPFF- and AR234958-bound Mas1 complexes

To elucidate the molecular basis of Mas1's adaptability to chemically diverse ligands, we compared the cryo-EM structures of Mas1 bound to the small-molecule agonist AR234958 (hot pink) and the neuropeptide NPFF (green) (Fig. 5A). Despite their stark differences in size, flexibility, and chemical composition—AR234958 as a compact synthetic molecule and NPFF as a flexible peptide—both ligands occupy highly similar spatial positions within the orthosteric binding pocket, extending from the extracellular vestibule to the receptor core (Fig. 5A,B).

Structural alignment reveals that the residues lining the binding cavity, primarily from TM2, TM3, TM6, and TM7, adopt nearly identical conformations in both complexes, indicating a pre-configured, robust pocket architecture (Fig. 5B,C). This conserved structural framework enables Mas1 to accommodate diverse ligands without significant receptor rearrangements, underscoring its ligand promiscuity. Residue H262[7.35] forms key

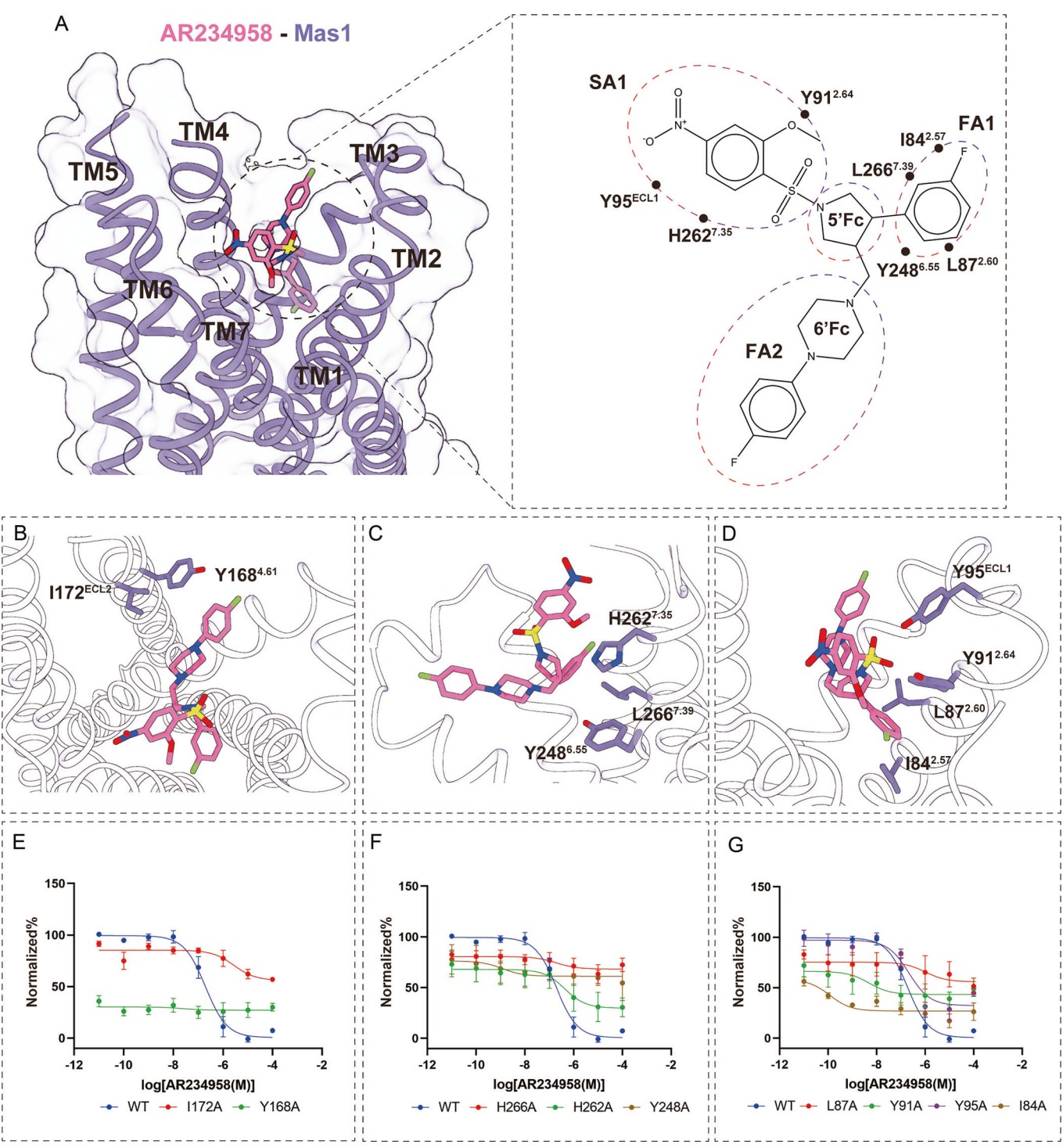

polar and aromatic interactions with the ligand (Fig. 5D,E), which contributes to shaping the binding pocket's intracellular boundary. Structure-guided mutagenesis revealed functional asymmetry: alanine substitution of H262[7.35] (H262A) severely impaired receptor activation, whereas H263A had minimal impact (Appendix Fig. S6a,b), establishing H262[7.35] as a critical anchor for ligand stabilization and setting the stage for subsequent activation steps.

## Hydrophobic relay-driven activation mechanism in Mas1

To unravel the molecular principles governing ligand-mediated activation in Mas1 and its MRGPR subfamily, we conducted comparative structural and functional analyses across multiple receptor states and related receptors (Fig. 6). Structural comparisons of ligand-bound Mas1 (NPFF, AR234958), MRGPRX1 (Liu et al, 2023), MRGPRX2 (Cao et al, 2021), and the class A aminergic

**Figure 4. Structural basis of AR234958 recognition by Mas1 and functional impact of key binding pocket residues.**

(A) Cross-sectional view of the Mas1 transmembrane domain (purple) with AR234958 (pink) occupying the orthosteric ligand-binding pocket. Right: schematic diagram illustrating the three binding subareas (SA1, FA1, FA2) of AR234958, which engages key residues via hydrophobic and aromatic interactions. Critical residues include I172$^{ECL2}$, Y168$^{4.61}$, Y248$^{6.55}$, H262$^{7.35}$, L266$^{7.39}$, L87$^{2.60}$, Y91$^{2.64}$, and Y95$^{ECL1}$. (B–D) Structural snapshots of specific residue interactions with AR234958. (B) The extracellular entrance. The FA2 moiety is oriented toward I172$^{ECL2}$ and Y168$^{4.61}$, which define the structural boundary of the binding pocket. (C) Central pocket coordination involving H262$^{7.35}$, Y248$^{6.55}$ and L266$^{7.39}$. (D) Lower cavity residues I84$^{2.57}$, Y95$^{ECL1}$, Y91$^{2.64}$, and L87$^{2.60}$ contribute to stabilizing the aromatic FA1 moiety and SA1 moiety of AR234958. (E–G) Concentration-dependent inhibition of cAMP accumulation in HEK293T cells expressing wild-type (WT) or mutant Mas1 variants. (E) I172A and Y168A mutations abrogate AR234958 activity. Data are presented as mean ± SEM from three independent experiments ($n = 3$). (F) Y248A, L266A and H262A significantly reduce ligand efficacy. Data are presented as mean ± SEM from three independent experiments ($n = 3$). (G) Alanine substitutions at I84, Y91, Y95, and L87 also attenuate ligand response. Data are presented as mean ± SEM from three independent experiments ($n = 3$). Source data are available online for this figure.

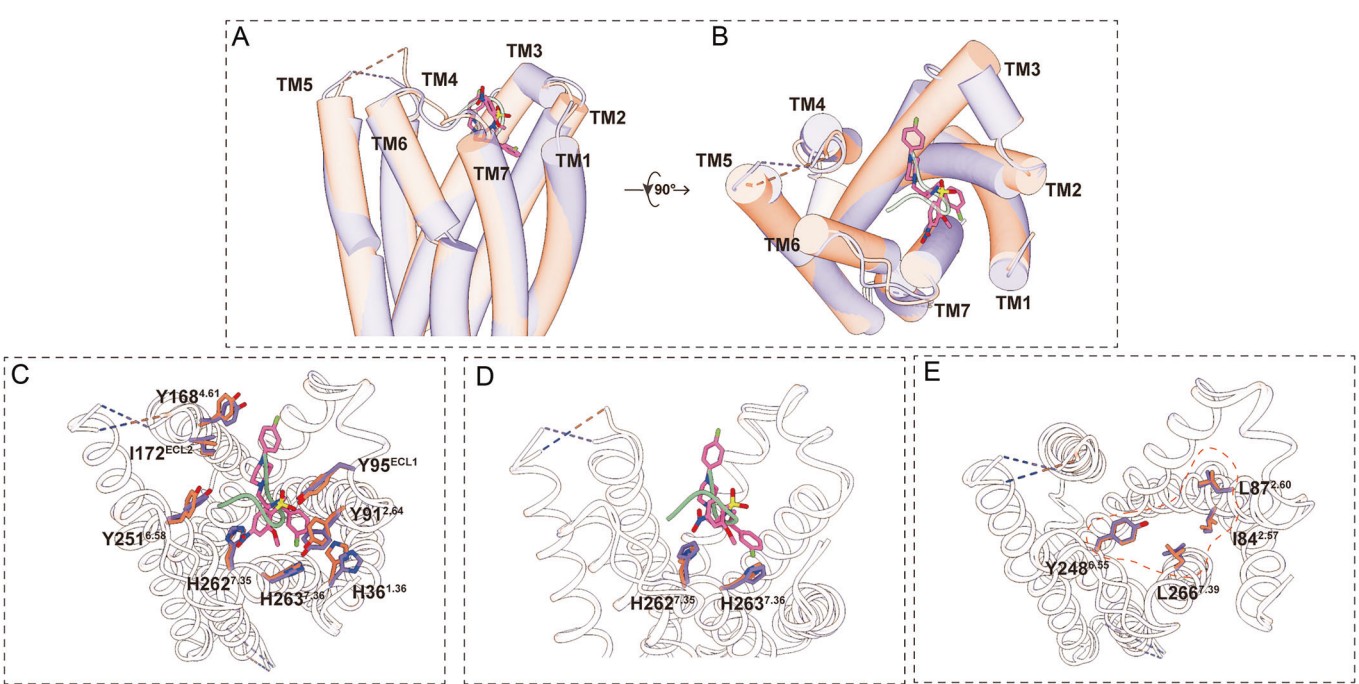

**Figure 5. Comparison of NPFF- and AR234958-bound mas1 structures reveals a conserved hydrophobic binding core and ligand-triggered conformational pressure switch.**

(A, B) Structural overlays of Mas1 bound to NPFF (green) or AR234958 (pink), viewed from the side (A) and the extracellular surface (B). Ligands occupy similar spatial positions within the transmembrane core, and induce local adjustments of helices TM2–TM6 to accommodate their distinct chemical scaffolds. (C) Both NPFF and AR234958 reside in a binding pocket formed by a similar set of conserved residues, including Y168$^{4.61}$, Y91$^{2.64}$, Y95$^{ECL1}$, I172$^{ECL2}$, Y251$^{6.58}$, H36$^{1.36}$, and H262$^{7.35}$. Despite ligand differences, these residues delineate a shared structural cavity supporting promiscuous yet selective binding. (D) Overlay of H262$^{7.35}$ and H263$^{7.36}$ shows their distinct conformational positions in the two complexes. Although both appear to contribute to ligand interaction structurally, complementary functional data (see Appendix Fig. S6) identify H262$^{7.35}$ as the principal hydrophobic anchoring residue essential for ligand-induced activation. (E) A conserved hydrophobic surface formed by Y248$^{6.55}$, L266$^{7.39}$, L87$^{2.60}$, and I84$^{2.57}$ is observed in both structures. These residues define a planar hydrophobic patch that likely serves as a conformational "pressure switch," stabilizing ligand binding through hydrophobic packing and enabling signal transduction upon engagement.

receptor D3R (PDB ID: 8IRT) revealed distinct mechanisms of engaging the conserved toggle switch residue W6.48, a hallmark of GPCR activation. In D3R (Xu et al, 2023), the ligand rotigotine directly contacts W6.48 at 3.5 Å, whereas NPFF- and AR234958-bound Mas1 complexes position their ligands at 12.7 Å and 14.9 Å, respectively, from this residue (Fig. 6A). Similarly, MRGPRX1 and MRGPRX2 maintain ligand distances of 13.1–18.0 Å from W6.48, indicating that Mas1 and its MRGPR relatives employ a non-canonical activation mechanism that bypasses direct W6.48 engagement, unlike classical class A GPCRs.

Comparison of active-state Mas1 structures with an AlphaFold3-predicted inactive-state model revealed a significant conformational shift upon activation, marked by a 16.8° outward displacement of TM6 (Fig. 6B). This movement is accompanied by localized rearrangements at the TM3–TM6 interface, where residues A241$^{6.48}$, F237$^{6.44}$, and L120$^{3.40}$ undergo lateral shifts of approximately 2.11, 4.37, and 4.98 Å, respectively. Top-down views show these residues also rotate along the horizontal axis, suggesting that activation involves coordinated axial and rotational adjustments within the helical bundle to facilitate G protein coupling (Fig. 6B). These rearrangements likely disrupt interhelical constraints between TM3 and TM6, enabling the structural transitions required for downstream signaling.

Based on these structural rearrangements, we propose a ligand-induced hydrophobic compression plane as a central feature of Mas1

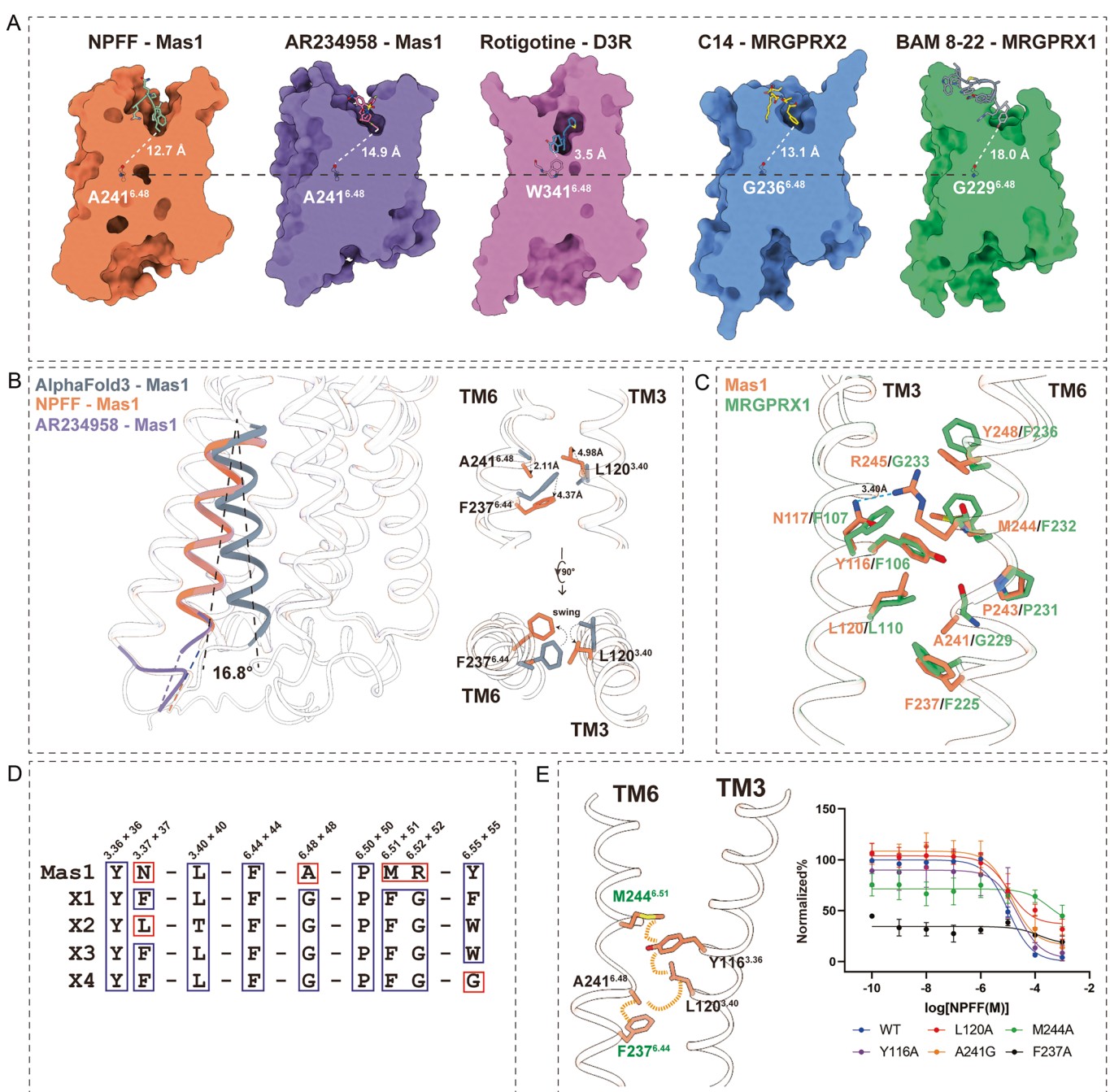

Figure 6. Comparative analysis of Mas1 and MRGPR family reveals conserved hydrophobic cleft and functional divergence in ligand recognition.

(A) Surface representations of the ligand-binding pockets in Mas1 (NPFF-bound and AR234958-bound), MRGPRX2 (C14-bound), MRGPRX1 (BAM8-22-bound), and dopamine receptor D3R (rotigotine-bound). Although Mas1 and MRGPRX family receptors share a conserved deep hydrophobic cavity shaped around position 6.48 (e.g., A241[6.48] in Mas1, G236[6.48] in MRGPRX2), their activation appears to be mediated indirectly through ligand engagement with residues surrounding the cavity, rather than via direct interaction with 6.48. In contrast, rotigotine in D3R makes direct contact with W341[6.48], exemplifying a canonical class A GPCR activation mechanism. (B) Left: Structural overlay of Mas1 in different states—AlphaFold3 inactive model (gray), NPFF (orange), and AR234958 (purple)—highlighting a 16.8° outward displacement of TM6 in ligand-bound states relative to AlphaFold3. Right: Side view of TM3–TM6 swing in Mas1 showing the reorientation of A241[6.48], L120[3.40], and F237[6.44]. (C) Structural comparison of Mas1 (orange) and MRGPRX1 (green) reveals conserved core residues within TM3–TM6, including Y116[3.36], M244[6.51], A241[6.48], and F237[6.44]. (D) Multiple sequence alignment of Mas1 with MRGPR paralogs (X1–X4), showing conservation at key positions along TM3 and TM6. Divergences (highlighted in red) may underlie differences in ligand selectivity and downstream coupling. (E) Functional validation of key TM3–TM6 pocket residues via cAMP accumulation assay in HEK293T cells expressing WT or mutant Mas1 receptors. Alanine substitutions at Y116[3.36], A241[6.48], M244[6.51], and F237[6.44] impair NPFF-induced signaling. Data are presented as mean ± SEM ($n = 3$). Source data are available online for this figure.

activation (Fig. 5E), which arises from cooperative interactions between the ligand and a network of hydrophobic residues, including Y248$^{6.55}$, L266$^{7.39}$, L87$^{2.60}$, and I84$^{2.57}$ forming a contiguous hydrophobic surface within the receptor core (Fig. 5E). We hypothesize that ligand binding compresses this plane, generating mechanical stress that propagates through the transmembrane helices to the intracellular interface, facilitating conformational changes essential for G protein coupling (Appendix Fig. S7). Mutagenesis of key plane residues, particularly H262$^{7.35}$ and L266$^{7.39}$, significantly reduced receptor activation, supporting this model.

Structural analysis of Mas1 and MRGPRX1 highlighted both shared and divergent features in their activation microswitch networks. Both receptors conserve hydrophobic residues at key positions, such as L120$^{3.40}$ and F237$^{6.44}$, suggesting a common TM3–TM6 coupling framework (Fig. 6C). However, Mas1 uniquely substitutes the canonical W6.48 toggle switch with an alanine (A241$^{6.48}$), likely altering the mechanical coupling pathway. This substitution, combined with variations in the surrounding microenvironment, underscores mechanistic diversity within the MRGPR subfamily. Sequence alignment with MRGPRX1–X4 further revealed low conservation of activation-associated residues, indicating that each receptor employs tailored structural elements for ligand-specific activation (Fig. 6D).

Mechanistic dissection of the active-state Mas1 structure identified a hydrophobic relay network driven by M244$^{6.51}$ and F237$^{6.44}$ as critical molecular switches (Fig. 6C). This network extends through Y116$^{3.36}$, L120$^{3.40}$, and A241$^{6.48}$, forming a contiguous hydrophobic plane that couples ligand binding to TM6 displacement. M244$^{6.51}$ initiates the relay by transmitting mechanical tension through A241$^{6.48}$, which shifts downward, driving the outward movement of F237$^{6.44}$ and promoting TM6 displacement. Functional validation via site-directed mutagenesis supported this model, as the A241G substitution, designed to disrupt the hydrophobic relay, significantly impaired receptor activation (Fig. 6E). These findings establish M244$^{6.51}$ and F237$^{6.44}$ as pivotal actuators of Mas1's non-canonical activation, operating through a hydrophobic relay that integrates ligand recognition with downstream signaling.

To contextualize the Mas1–Gi interface, we superposed the Mas1–Gi complex with Gi-bound structures of MRGPRX1 (PDB ID: 8JGG) and NTSR1 (PDB ID: 7L0P) (Zhang et al, 2021) (Appendix Fig. S8). The active 7TM scaffold and the insertion of the Gαi C-terminal α5 helix are overall conserved; in Mas1, α5 shows only a slight rotation toward TM7/H8 relative to the references. By contrast, the prevailing difference resides in the Gαi αN helix, which is laterally displaced and subtly reoriented relative to MRGPRX1 and NTSR1 (A11 Cα separations of ~10.3 Å and ~15.6 Å across pairwise overlays; Appendix Fig. S8). These observations indicate that the distinct geometry of the Mas1–Gi complex is dominated by αN repositioning rather than changes in α5 engagement.

## Discussion

The cryo-EM structures of Mas1 bound to neuropeptide FF (NPFF) and the synthetic small-molecule AR234958 provide a transformative framework for understanding the molecular basis of ligand recognition and activation in this therapeutically significant G protein-coupled receptor (GPCR). Our findings reveal a conserved, hydrophobic orthosteric binding pocket that accommodates chemically diverse ligands through shared aromatic and hydrophobic interactions, underscoring Mas1's remarkable structural plasticity. This promiscuous yet selective binding architecture enables Mas1 to engage a range of agonists, from peptides like NPFF to small molecules like AR234958, facilitating its diverse physiological roles in cardiovascular homeostasis, inflammation suppression, and neuroendocrine regulation.

Notably, our dose–response analyses indicate that angiotensin-(1–7) (Ang-(1–7)), previously considered a potential endogenous ligand for Mas1, exhibits a low potency with an EC50 in the millimolar range. This significantly reduced potency compared to NPFF and AR234958, which achieve half-maximal activation at much lower concentrations, suggests that Ang-(1–7) is unlikely to serve as a physiologically relevant ligand for Mas1 under typical conditions. Instead, ligands like NPFF and AR234958, with their higher potency and structural compatibility with the receptor's binding pocket, appear better suited to drive Mas1-mediated signaling.

A pivotal discovery of this study is the identification of a non-canonical activation mechanism in Mas1, which diverges from the classical GPCR paradigm reliant on direct engagement of the W6.48 toggle switch. Mas1 employs a ligand-induced hydrophobic compression plane, comprising residues Y248$^{6.55}$, L87$^{2.60}$, I84$^{2.57}$, and L266$^{7.39}$, to transduce mechanical stress from the orthosteric pocket to the intracellular interface. This plane, coupled with a hydrophobic relay network centered on M244$^{6.51}$ and F237$^{6.44}$, drives the outward displacement of TM6, enabling G protein coupling. Functional mutagenesis validates the critical roles of these residues, as their disruption significantly impairs receptor activation. This mechanistic divergence distinguishes Mas1 from canonical class A GPCRs and positions it as a model for studying alternative GPCR activation strategies within the MRGPR subfamily.

Comparative analyses with MRGPRX1–X4 highlight both conserved and divergent features within this receptor family. While the hydrophobic core of the binding pocket is preserved, sequence variability at key activation-associated residues suggests receptor-specific adaptations that tailor ligand selectivity and signaling outcomes. These findings illuminate the evolutionary flexibility of the Mas1/MRGPR family, revealing how structural variations underpin functional specialization in response to distinct physiological demands.

The implications of our work are profound for both basic science and translational research. By elucidating the structural determinants of Mas1's ligand promiscuity and activation, we provide a molecular blueprint for rational drug design targeting this receptor. Mas1's roles in mitigating cardiovascular dysfunction, inflammation, and metabolic disorders position it as a promising therapeutic target for conditions such as hypertension, heart failure, and chronic inflammation. The high-resolution insights into Mas1's binding pocket and activation machinery enable the development of selective agonists that could enhance its protective effects, offering novel interventions for these high-burden diseases.

Furthermore, our study advances the broader field of GPCR structural biology by demonstrating how non-canonical activation mechanisms achieve signaling specificity. The hydrophobic compression plane and relay network identified here may serve as a paradigm for other orphan or understudied GPCRs, guiding future investigations into their pharmacological modulation. Collectively, these findings deepen our understanding of Mas1 pharmacology

and establish a foundation for precision therapeutics that harness its signaling pathways, paving the way for innovative treatments in complex diseases.

# Methods

### Reagents and tools table

| Reagent/resource | Reference or source | Identifier or catalog number |
| --- | --- | --- |
| **Experimental models** | | |
| *E. coli* OmniMAX 2 T1 | Invitrogen | Cat# C854003 |
| *E. coli* DH10B | Invitrogen | Cat# EC0113 |
| Spodoptera frugiperda (Sf9) | Expression systems | Cat#94-001 F |
| HEK293T | ATCC | |
| **Recombinant DNA** | | |
| pFastBac BRIL-Mas1 | This study | |
| pcDNA6 Mas1 | This study | N/A |
| **Antibodies** | | |
| N/A | N/A | N/A |
| **Oligonucleotides and other sequence-based reagents** | | |
| PCR primers | Tsingke | |
| **Chemicals, enzymes and other reagents** | | |
| AR234958 | Chemical Department | Synthesis |
| NPFF | tgpeptide | Synthesis |
| Ang 1–7 | TargetMol | Cat# T7399 |
| ESF921 insect cell culture medium | Expression system | Cat# 96-001-20 |
| Protease Inhibitor Cocktail | TargetMol | Cat# C0001 |
| Lauryl Maltose Neopentyl Glycol | Anatrace | Cat# NG310 |
| Cholesteryl Hemisuccinate Tris Salt | Anatrace | Cat# CH210 |
| Glyco-diosgenin | Anatrace | Cat# GDN101 |
| Dulbecco's Modified Eagle Medium (DMEM) | Cytiva | Cat# SH30243.01 |
| Fetal Bovine Serum | ExCell | Cat# FSP500 |
| Lipofectamine 2000 | Invitrogen | Cat# 11668019 |
| Opti-MEM | Gibco | Cat# 11058021 |
| ClonExpress II One Step Cloning Kit | Vazyme | Cat#112-01/02 |
| Apyrase | Sigma | Cat#9000-95-7 |
| TALON Metal Affinity Resin | TAKARA | Cat#635501 |
| **Software** | | |
| CryoSPARC | https://cryosparc.com/ | Version 4.5.1 |
| MotionCor2 | https://msg.ucsf.edu/software | Version 1.6.4 |
| PHENIX | https://phenix-online.org/ | Version 1.21.1-5286 |
| COOT | www.2.mrc-lmb.cam.ac.uk/personal/pemsley/coot/ | Version 0.9.8.93 |
| UCSF Chimera | https://www.cgl.ucsf.edu/chimera/ | Version 1.17.3 |
| UCSF ChimeraX | https://www.cgl.ucsf.edu/chimerax/ | Version 1.7.1 |
| PyMOL | https://pymol.org/ | Version 2.5 |
| GraphPad Prism | https://www.graphpad.com/scientific-software/prism/ | Version 8.0.1 |
| AlphaFold3 | https://alphafoldserver.com | |
| **Other** | | |
| EnVision plate reader | PerkinElmer | |
| BD BiosciencesFlow Cytometer | BD | |

## Constructs

The full-length human Mas1 (residues 1–325) was cloned into the pFastBac (Thermo Fisher Scientific) vectors using the ClonExpres-sIIOne Step Cloning Kit (Vazyme Biotech), along with a His-Tag consisting of 6 consecutive histidines and the N-terminal haemagglu-tinin (HA) signal peptide. Additionally, a tobacco etch virus (TEV) cleavage site was introduced between the N-terminus and bRIL fusion protein for complex purification. The engineered $G_{\alpha i}$ chimeric proteins were designed as chimeras based on the mini-$G_{s/i1}43$ skeletons (Li et al, 2024; Nehmé et al, 2017), respectively, with the N-terminal 1–18 amino acids and the $G\alpha AH$ domain of $G_{i1}$ replaced to facilitate binding to scFv16 (Kim et al, 2020; Li et al, 2024; Nehmé et al, 2017; Wang et al, 2021; Yin et al, 2021). Human wild-type (WT) $G_{\beta1}$, human $G_{\gamma2}$, and scFv16 were cloned into pFastBac vectors.

## Insect cell expression

The WT human Mas1, $G_i$ chimera, $G_{\beta1}$, $G_\gamma$, and scFv16 were co-expressed in Sf9 insect cells using the baculovirus method (Promega). Cell cultures were grown in ESF 921 serum-free medium (Expression Systems) to a density of 3.5 million cells per mL and then infected with six individual baculoviruses at a suitable ratio. The culture was collected by centrifugation 48 h after infection, and cell pellets were stored at −80 °C.

## Complex purification

Cell pellets were thawed in 20 mM HEPES pH 7.4, 100 mM NaCl, 5 mM $MgCl_2$, and $CaCl_2$ supplemented with Protease Inhibitor Cocktail (TargetMol). For the NPFF-Mas1-$G_i$-scFv16 complex, 100 µM NPFF (TGpeptide) and 25 mU ml$^{-1}$ apyrase (Sigma) were added. For the AR234958-Mas1-$G_i$-scFv16 complex, 100 µM AR234958 (Synthesis) and 25 mU ml$^{-1}$ apyrase (Sigma) were added. The suspension was incubated for 30 min at room temperature, and the complex was solubilized from the membrane using 0.5% (w/v) lauryl maltose neopentylglycol (LMNG) (Ana-trace) and 0.1% (w/v) cholesteryl hemisuccinate (CHS) (Anatrace) for 3 h at 4 °C. Insoluble material was removed by centrifugation at

65,000× *g* for 30 min, and the supernatant was purified by His-Tag affinity chromatography (TALON Metal Affinity Resin, TAKARA). The resin was then packed and washed with 40 column volumes of 20 mM HEPES pH 7.4, 100 mM NaCl, 0.05% (w/v) LMNG, 0.01% CHS, and 50 μM ligand, 20 mM Imidazole. The complex sample was eluted in buffer containing 20 mM HEPES pH 7.4, 100 mM NaCl, 0.01% (w/v) LMNG, 0.002% CHS, 20 μM ligand, 200 mM Imidazole. Complex fractions were concentrated with a 100-kDa molecular weight cut-off (MWCO) Millipore concentrator for further purification. The complex was then subjected to size-exclusion chromatography on a Superdex 200 Increase 10/300 GL column (Cytiva) pre-equilibrated with size buffer containing 20 mM HEPES pH 7.4, 100 mM NaCl, 0.00075% (w/v) LMNG, 0.00025% (w/v) GDN (Anatrace), 0.00015% CHS, and 10 μM ligand to separate complexes. Eluted fractions were evaluated by SDS-PAGE, and those consisting of receptor-$G_i$ protein complexes were pooled and concentrated for cryo-EM experiments.

## Cryo-EM grid preparation and data acquisition

Two microliters of the purified complexes at around, 16 mg ml$^{-1}$ and 19 mg ml$^{-1}$ for NPFF-Mas1-$G_i$ complexes and AR234958-Mas1-$G_i$ complexes, respectively, were applied onto a glow-discharged Quantifoil Au 1.2/1.3 300-mesh gold holey carbon grid. The sample was incubated for 5 s on the grids before blotting for 3 s under 100% humidity at 4 °C and then vitrified by plunging into liquid ethane using a Vitrobot Mark IV (Thermo Fisher Scientific). Cryo-EM data collection was performed at the Advanced Center for Electron Microscopy, Shanghai Institute of Materia Medica. A total of 5,150/14,212 movies for the NPFF-Mas1-$G_i$/AR234958-Mas1-$G_i$ complexes were collected on Gatan K3 direct electron detection device. Images were taken with a pixel size of 0.83 Å, a defocus ranging from −1.0 to −2.0 μm, and total dose of 50 e Å$^2$ s$^{-1}$, using the EPU software (FEI Eindhoven, Netherlands).

## Cryo-EM data processing

Single-particle cryo-EM analysis of the Mas1-$G_i$ complex was conducted using cryoSPARC v4 (Punjani et al, 2017). Dose-fractionated image stacks underwent motion correction via MotionCor2 (Zheng et al, 2017), and the contrast transfer function (CTF) parameters were estimated using the patch-based CTF estimation approach.

For the NPFF-Mas1-$G_i$ complexes, particle picking was carried out automatically from a down-sampled dataset (pixel size 1.66 Å), followed by unsupervised 2D classification to discard particles with poor structural features. After two iterative rounds of 2D classification, 1,734,471 high-quality particles were retained. These were subjected to ab initio 3D reconstruction and heterogeneous refinement, after which two particle subsets were re-extracted at full resolution (pixel size 0.83 Å). A total of 1,287,698 particles were used for subsequent 3D classification. Among these, four subsets comprising 782,093 particles were selected based on superior ligand density quality, and were further refined using non-uniform refinement and local refinement strategies. The final 3D reconstruction yielded a density map at a global resolution of 2.54 Å, as determined by the gold-standard Fourier shell correlation (FSC) at the 0.143 threshold. Additional focused refinement on the receptor region produced a Mas1-focused map with a resolution of 2.77 Å.

For the AR234958-Mas1-$G_i$ complexes, particle picking was carried out automatically from a down-sampled dataset (pixel size 1.66 Å), followed by unsupervised 2D classification to discard particles with poor structural features. After two iterative rounds of 2D classification, 1,144,380 high-quality particles were retained. These were subjected to ab initio 3D reconstruction and heterogeneous refinement, after which two particle subsets were re-extracted at full resolution (pixel size 0.83 Å). A total of 502,230 particles were picked as reference for particle picking. Particles with good performance are picked through interactive rounds of 2D and 3D classifications. The final 162,204 particles were further refined using non-uniform refinement and local refinement strategies. The final 3D reconstruction yielded a density map at a global resolution of 3.07 Å, as determined by the gold-standard Fourier shell correlation (FSC) at the 0.143 threshold. Additional focused refinement on the receptor region produced a Mas1-focused map with a resolution of 3.31 Å.

## Model building and refinement

For the Mas1-$G_i$ complexes, the AlphaFold3 structure of Mas1 (Jumper et al, 2021), and the structure of $G_i$ complex (PDB code: 8ZPS) (Li et al, 2024) were used as the initial model for model rebuilding and refinement against the electron microscopy map. The models of Mas1 were built according to the Mas1-focused maps and Mas1-$G_i$ complexes are building according to the full maps. The model was docked into the electron microscopy density map using Chimera (Pettersen et al, 2004), followed by iterative manual adjustment and rebuilding in COOT and ISOLDE (Croll, 2018; Emsley and Cowtan, 2004). Real space and reciprocal space refinements were performed using Phenix (Adams et al, 2010) programs. The model statistics were validated using MolProbity (Chen et al, 2010). The final refinement statistics were validated using the module "comprehensive validation (cryo-EM)" in Phenix. The final refinement statistics are provided in Appendix Table S4. Structural figures were prepared in ChimeraX (Pettersen et al, 2021) and PyMOL (https://pymol.org/2/) (Appendix Table S4).

## cAMP accumulation assay

cAMP accumulation was measured by a LANCE Ultra cAMP kit (PerkinElmer) according to the manufacturer's instructions. WT or mutant Mas1 constructs were cloned into pcDNA6 vector (Invitrogen) for functional studies(Appendix Tables S2 and S3). HEK293T cells were transiently transfected with the vectors using Lipofectamine 3000 transfection reagent (Invitrogen) and incubated at 37 °C in 5% CO$_2$. After 24 h, the transfected cells were digested with 0.02% (w/v) EDTA, resuspended in stimulation buffer (Hanks' balanced salt solution (HBSS)) supplemented with 5 mM HEPES, 0.5 mM 3-isobutyl-1-methylxanthine and 0.1% (w/v) bovine serum albumin (BSA), pH 7.4) and added into 384-well white plates (PerkinElmer) with a density of 6 × 10$^5$ cells/mL. The cells were stimulated with different concentrations of ligands plus 800 nM forskolin and were incubated for 40 min at room temperature. The reaction was stopped by adding 5 μL Eu-cAMP tracer and 5 μL ULight-anti-cAMP working solution separately. After one hour additional incubation, the plates were read in an EnVision multilabel plate reader (PerkinElmer) to measure time-resolved fluorescence resonance energy transfer (TR-FRET) signals at 620 and 665 nm. Data were analyzed in GraphPad Prism 8.

## Receptor surface expression

Cell surface expression was determined by flow cytometry to detect the N-terminal Flag tag on the WT Mas1 (pcDNA6-3×Flag-Mas1) and its mutant constructs transiently expressed in HEK293T cells. Briefly, approximately $2 \times 10^5$ cells were blocked with PBS containing 5% BSA (w/v) at room temperature for 15 min, and then incubated with 1:300 anti-Flag primary antibody (diluted with PBS containing 5% BSA, Sigma-Aldrich, Cat#F3165, purified IgG1 subclass) at room temperature for 60 min. The cells were then washed three times with PBS containing 1% BSA (w/v) followed by 60 min incubation with 1:1000 anti-mouse Alexa Fluor 488 conjugated secondary antibody (diluted with PBS containing 5% BSA, Invitrogen, Cat#A-21202) at room temperature in the dark. After washing three times, cells were resuspended in 200 μL PBS containing 1% BSA for detection by Flow Cytometer (BD Biosciences) utilizing laser excitation and emission wavelengths of 488 and 530 nm, respectively. For each sample, 10,000 cellular events were collected, and the total fluorescence intensity of the positive expression cell population was calculated. Data were normalized to the WT receptor.

## Statistics

All functional study data were analyzed using GraphPad Prism 8.0 (Graphpad Software Inc.) and showed as means ± S.E.M. from at least three independent experiments in triplicate. The significance was determined by one-way ANOVA with Dunnett's multiple comparisons test, and $*P < 0.05$ was considered statistically significant.

# Data availability

The coordinates and map density data from this publication have been deposited to the RCSB Protein Data Bank (RCSB PDB: http://www.rcsb.org) and The Electron Microscopy Data Bank (EMDB: http://www.ebi.ac.uk/emdb) and assigned the identifier as 9X3Y (NPFF-Mas1-local refinement), 9X3Z (NPFF-Mas1-Gi complex), 9X41 (AR234958-Mas1-local refinement) and 9X40 (AR234958-Mas1-Gi complex).

The source data of this paper are collected in the following database record: biostudies:S-SCDT-10_1038-S44318-026-00764-6.

# Peer review information

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

## Acknowledgements

The cryo-EM data were collected at the Advanced Center for Electron Microscopy at the Shanghai Institute of Materia Medica, Chinese Academy of Sciences. We are grateful to Kai Wu for collecting the cryo-EM data. This work was supported by the National Natural Science Foundation of China (32130022, 82495184, and 82121005 to HEX; 82473981, 82273985, and 82121005 to DHY); the National Key R&D Program (2022YFC2703105 to HEX; 2023YFA1800804 to DHY); the National Key R&D Program "Strategic Scientific and Technological Innovation Cooperation" Key Project (2022YFE0203600) released by the Ministry of Science and Technology; the CAS Strategic Priority Research Program (XDB37030103 and XDB0830000 to HEX; XDB1060402 to DHY); the Shanghai Municipal Science and Technology Major Project (2019SHZDZX02 to HEX); the Lingang Laboratory (LGL-2612-05 to DHY); and the National Science and Technology Major Project of China (2025ZD1802003 to DHY).

## Author contributions

**Yumu Zhang**: Conceptualization; Data curation; Formal analysis; Supervision; Validation; Investigation; Visualization; Methodology; Writing—original draft; Project administration; Writing—review and editing. **Qiuying Wang**: Formal analysis; Investigation; Writing—review and editing. **Heng Liu**: Formal analysis; Investigation; Writing—review and editing. **Hong Shan**: Formal analysis; Validation. **Yimin Gu**: Writing—review and editing. **Jiaqi Yang**: Investigation. **Yuan Gao**: Writing—review and editing. **Kai Wu**: EM data Collection. **Dehua Yang**: Resources; Supervision; Funding acquisition; Investigation; Project administration. **H Eric Xu**: Conceptualization; Resources; Supervision; Funding acquisition; Project administration.

Source data underlying figure panels in this paper may have individual authorship assigned. Where available, figure panel/source data authorship is listed in the following database record: biostudies:S-SCDT-10_1038-S44318-026-00764-6.

## Disclosure and competing interests statement

The authors declare no competing interests.

