## [Peer Review File · The EMBO Journal]

Structural Insight into Ligand Binding and Activation of the Orphan GPCR Mas1

Yumu Zhang, Qiuying Wang, Heng Liu, Hong Shan, Yimin Gu, Jiaqi Yang, Yuan Gao, Kai Wu, Dehua Yang, and Huaqiang Xu

Corresponding author(s): Huaqiang Xu (eric.xu@simm.ac.cn), Dehua Yang (dhyang@simm.ac.cn), Yumu Zhang (yumu_zhang@hms.harvard.edu)

Review Timeline:

Submission Date:	25th Jun 25
Editorial Decision:	23rd Jul 25
Revision Received:	28th Sep 25
Editorial Decision:	6th Nov 25
Revision Received:	3rd Mar 26
Accepted:	10th Mar 26

Editor: Hartmut Vodermaier

Transaction Report:

Dr. Huaqiang Eric Xu
Shanghai Institute of Materia Medica
Chinese Academy of Sciences
Shanghai
China

23rd Jul 2025

Re: EMBOJ-2025-121709
Structural Insight into Ligand Binding and Activation of the GPCR Orphan Receptor Mas1

Dear Dr. Xu,

Thank you again for submitting your study on Mas1 GPCR structure to The EMBO Journal. I sent it to three expert referees, whose comments are copied below for your information. As you will see, all referees are generally supportive of publication. However, while referees 1 and 2 request for the most part only presentational changes, referee 3 raises more major concerns regarding the structural analyses. In particular, this referee is not convinced that the model building based on the provided electron density maps is correct for one structure, something s/he is trying to illustrate in the attached image and structure coordinate files, associated with the suggestion to rebuild the model in a different way.

In light of these reports and recommendations, we would be interested in pursuing a revised version of the study further for EMBO Journal publication, pending adequate addressing of the points raised in the three reports. Please note that it is our policy to consider only a single round of major revision, making it important to fully respond to all comments at the time of resubmission; therefore, please do not hesitate to get back to me in case you would like to clarify/discuss any of the referees' points or plans for addressing them ahead of time. We would also be open to extending the revision deadline if that should be helpful.

Detailed information on preparing, formatting and uploading a revised manuscript can be found below and in our Guide to Authors. Thank you again for the opportunity to consider this work for The EMBO Journal, and I look forward to your revision in due time.

With kind regards,

Hartmut

9) To facilitate reproducibility and cross-laboratory adoption of methodologies, please structure the Materials & Methods section as outlined in our guide to authors, including a completed Reagents and Tools Table that can be downloaded from our author guidelines as well (<https://www.embopress.org/page/journal/14602075/authorguide#structuredmethods>).

10) Digital image enhancement is acceptable practice, as long as it accurately represents the original data and conforms to community standards. If a figure has been subjected to significant electronic manipulation, this must be clearly noted in the figure legend and/or the 'Materials and Methods' section. The editors reserve the right to request original versions of figures and the original images that were used to assemble the figure. Finally, we generally encourage uploading of numerical as well as gel/blot image source data; for details see: embopress.org/page/journal/14602075/authorguide#sourcedata

In the interest of ensuring the conceptual advance provided by the work, we recommend submitting a revision within 3 months (21st Oct 2025). Please discuss the revision progress ahead of this time with the editor if you require more time to complete the revisions. Use the link below to submit your revision:

Link Not Available

Referee #1:

The authors solved cryo-EM structures of Mas1 activated by natural peptide NPFF and synthetic molecule AR234958 in complex with a chimeric Gi. The structures reveal the basis of Mas1 activation by disparate agonists, the only common feature of which is hydrophobicity. Structural evidence of the interactions of ligands with Mas1 was confirmed by receptor mutagenesis. The structures also suggest a mechanism of receptor activation distinct from many class A GPCRs, which is the major achievement of this study. Overall, the study is rigorous, and the results are important for the GPCR field. However, several changes in the presentation, particularly the addition of info on Gi coupling, are necessary.

1. Lines 84-86. The authors say "emerging evidence" while citing a 2006 paper. Correct.

2. The authors treat AlphaFold3 predicted structure of inactive Mas1 as data. Predictions are not data, they must be validated (or debunked) by experiments. Correct.

3. The authors use Ballesteros-Weinstein nomenclature. They should explicitly state this and reference the paper where this nomenclature was proposed.

4. The authors should add some discussion of the Mas1 interface with Gi, comparing it with solved structures of other Gi-coupled GPCRs in the results.

5. NPFF is a natural peptide, whereas AR234958 is a synthetic molecule. Based on the data, the authors correctly argue that Ang-(1-7) is unlikely be an endogenous agonist for Mas1, but this makes NPFF a possible candidate, not AR234958 that is not endogenous.

6. Fig.1. Panel a is not informative and should be eliminated. Panel b should include the structure of AR234958, indicating which peptide residues and AR234958 moieties mediate Mas1 binding.

7. Fig. 2. Ligands in panels a and b should be oriented similarly, with key hydrophobic elements involved in Mas1 binding facing the same way.

Referee #2:

This is an interesting paper on the Mas1 receptor--which is long-awaited for the field. I have only a few minor concerns:
SPECIFIC COMMENTS:

1. For the various mutants it is not clear that damaging mutations did not affect cell surface expression--something easy to quantify.
2. The authors did not cite key refs regarding MRGPRX1 structure and function: PMID: 36302898

Referee #3:

The manuscript reports the first high-resolution structure of MAS1 in complex with two distinct agonists, NPFF and AS234958, laying a structural foundation for rational drug development targeting this important receptor.

However, I have significant concerns regarding the quality of the density maps and the accuracy of the structural modeling--particularly for the AR234958-bound structure. Given that these structures may be used to guide drug discovery efforts, it is crucial that the models are built correctly.

The ligand density is suboptimal in both complexes, although this is a known challenge in GPCR-G protein structures, particularly in the upper portion of the receptor where flexibility often leads to poor map quality. In the case of NPFF, the overall map is reasonable, except for residues Q14 and P15. I recommend that the authors perform focused 3D classifications to explore potential conformational heterogeneity in the peptide. Considering the large particle number (>782,000), there is a good chance of identifying a subset with improved density for the NPFF-bound state. That said, even without further improvements, the current NPFF model is acceptable.

My primary concern lies with the AR234958-bound structure (PDB: sm-fitJ340-coot-18; map: P210_J340FLIP.mrc). First, there appears to be visible density for residues 96-98, yet they are not modeled. Given that this loop is near the ligand-binding pocket, accurate modeling is essential. Additionally, residues 255-259 do not seem to fit the map well. Most critically, the ligand modeling itself raises concerns. As shown in the figures, there are discontinuities and extra densities when comparing the ligand model to the map.

I acknowledge the difficulty in modeling due to the limited map quality. Again, focused 3D classification might help improve the local resolution. I attempted to build an alternative model for AR234958 using the current map, and while it is not perfect, certain aspects appear improved compared to the current model (see attached file). The authors need to provide strong evidence that their current model is the most accurate interpretation of the data.

I have a few additional comments regarding the signaling assays.

1. If I understand correctly, at very low ligand concentrations, the cAMP levels are primarily influenced by forskolin. In other words, the dose-response curves for each mutant should ideally start from a similar baseline. If this is the case, I suggest that the authors re-normalize the curves for better comparability.
2. It is unclear why the L87A mutant in Figure 4 shows an increase in signal. The authors should provide a plausible explanation for this observation.
3. The title of Supplementary Figure 6 is misleading. It should not be referred to as "AF3-bound MAS1," but rather as "AF3-predicted MAS1 structure."

Response to Reviewers – EMBO Journal Submission**Meta Information****Manuscript Title: Structural Insight into Ligand Binding and Activation of the GPCR Orphan Receptor Mas1****Manuscript ID: Manuscript EMBOJ-2025-121709****Corresponding Author: Yumu Zhang, Dehua Yang, H.Eric Xu****Revision Round: Round 1****Submission Date: 20250928****In this revised version, we have carefully addressed all reviewer comments:**

- ©Revised wording and citations to improve accuracy and clarity.
- ©Clarified the use of AlphaFold3-predicted models, explicitly distinguishing them from experimental data.
- ©Expanded discussion of the Mas1–Gi interface in the context of other Gi-coupled GPCRs.
- ©Added flow cytometry assays to confirm mutant surface expression levels.
- ©Included additional references and corrected figure labeling/orientations for consistency and clarity.
- ©Rebuilt and refined the AR234958–Mas1 model following the reviewer's suggestions, supported by energetic analyses, functional data, and improved map fitting.

We believe that these revisions substantially improve the rigor, accuracy, and clarity of our work. We are grateful to the reviewers for their insightful guidance, which has helped us strengthen the manuscript.

Reviewer #1 – Comment 1

Lines 84-86. The authors say "emerging evidence" while citing a 2006 paper.

Response: We thank the reviewer for pointing out this inconsistency. The phrase “emerging evidence” was indeed inappropriate when referring to a paper published in 2006. We have revised the text to read “multiple evidence” instead of “emerging evidence.” And we have corrected the citation as below.

...However, multiple evidence indicates that Mas1 can also couple to Gαq/11 and Gα12/13 proteins in specific cellular contexts, triggering intracellular calcium flux, RhoA activation, and cytoskeletal reorganization(Chikumi, Vázquez-Prado et al., 2002, Gavard & Gutkind, 2008, Sureshkumar, Souza Dos Santos et al., 2023).

Reviewer #1 – Comment 2

The authors treat AlphaFold3 predicted structure of inactive Mas1 as data. Predictions are not data, they must be validated (or debunked) by experiments.

Response: We thank the reviewer for this important comment. We agree that AF3 predictions cannot be considered as experimental data. In our study, the AF3-predicted MAS1 structure was only used as a computational reference model for comparison with the experimental cryo-EM structures, not as evidence. This practice is consistent with many recent GPCR studies where AF2/3-predicted structures are used as inactive-like references for structural comparison.

The cited article are listed below:

- 1.Mechanisms of ligand recognition and activation of melanin-concentrating hormone receptors. Cell Discovery 10: 48
 - 2.Molecular basis of ligand recognition and activation of the human succinate receptor SUCR1. Cell Research 34: 594-596
 - 3.Structural and functional evidence that GPR30 is not a direct estrogen receptor. Cell Research 34: 530-533
- (e.g., (He, Yuan et al., 2024, Li, Liu et al., 2024, Liu, Guo et al., 2024)

Furthermore, we compared the AF3-predicted MAS1 model with multiple experimentally determined inactive-state structures of other Class A GPCRs. We observed a high degree of structural similarity, supporting the notion that AF3-predicted ligand-free models indeed capture key features of the inactive conformation. We have revised the text accordingly to avoid any misinterpretation. We noticed that our original labeling was misleading in the title of Appendix Figure S6, and we have corrected the figure title.

AF3-Mas1 vs inactive MOR

AF3-Mas1 vs inactive BLT1

AF3-Mas1 vs inactive DP1

Reviewer #1 – Comment 3

The authors use Ballesteros-Weinstein nomenclature. They should explicitly state this and reference the paper where this nomenclature was proposed.

Response: We thank the reviewer for pointing this out. We agree that it is important to clearly state the nomenclature system. In the revised manuscript, we now explicitly mention that the Ballesteros–Weinstein numbering scheme was used throughout to annotate transmembrane (TM) residues, and we have cited the original reference as below

“...(Ballesteros-Weinstein nomenclature(Ballesteros & Weinstein, 1995))”

Reviewer #1 – Comment 4

The authors should add some discussion of the Mas1 interface with Gi, comparing it with solved structures of other Gi-coupled GPCRs in the results.

Response:We thank the reviewer for this valuable suggestion. In the revised manuscript, we have expanded our discussion of the Mas1–Gi interface. Specifically, we compared the interactions between Mas1 and Gi with those observed in other Gi-coupled GPCR structures, such as MRGPRX1 and the NTSR1. We found that while the overall interaction mode between the C-terminal $\alpha 5$ helix of Gi and the receptor core is conserved, the main difference is in the Gai αN helix. The sentence which include the comparison is added as below, and we added the Appendix Figure S7 as the expanded:

To contextualize the Mas1–Gi interface, we superposed the Mas1–Gi complex with Gi-bound structures of MRGPRX1(PDB ID: 8JGG)(Guo, Zhang et al., 2023) and NTSR1(PDB ID: 8JGG)(Zhang, Gui et al., 2021)(Appendix Figure S7). The active 7TM scaffold and the insertion of the Gai C-terminal $\alpha 5$ helix are overall conserved; in Mas1, $\alpha 5$ shows only a slight rotation toward TM7/H8 relative to the references. By contrast, the prevailing difference resides in the Gai αN helix, which is laterally displaced and subtly reoriented relative to MRGPRX1 and NTSR1 (A11 C α separations of ~ 10.3 Å and ~ 15.6 Å across pairwise overlays; Appendix Figure S7). These observations indicate that the distinct geometry of the Mas1–Gi complex is dominated by αN repositioning rather than changes in $\alpha 5$ engagement.

Reviewer #1 – Comment 5

NPFF is a natural peptide, whereas AR234958 is a synthetic molecule. Based on the data, the authors correctly argue that Ang-(1-7) is unlikely be an endogenous agonist for Mas1, but this makes NPFF a possible candidate, not AR234958 that is not endogenous.

Response: We thank the reviewer for this important comment. We agree that AR234958 is a synthetic tool compound and should not be considered as an endogenous ligand. Regarding NPFF, while our structural and functional data demonstrate that NPFF can engage and activate Mas1, we currently have no direct evidence that its physiological distribution overlaps with Mas1 expression, nor that it functions as the true endogenous ligand in vivo. Therefore, we have revised the text to more cautiously state that NPFF is a potential ligand for Mas1 but that its endogenous role remains to be firmly established.

Reviewer #1 – Comment 6

Fig.1. Panel a is not informative and should be eliminated. Panel b should include the structure of AR234958, indicating which peptide residues and AR234958 moieties mediate Mas1 binding.

Response: We thank the reviewer for the helpful suggestion. We have removed the original Fig. 1a and updated all in-text references to reflect the new panel lettering for Fig. 1. Regarding the request to add the AR234958 structure to Fig. 1b, we respectfully maintain that presenting the AR234958 2D structure and its annotated moieties in Fig. 4 is more consistent with the narrative flow of the manuscript, where we analyze the Mas1 binding pocket and report the corresponding mutational data. In the revised Fig. 4, the AR234958 moieties are shown alongside the Mas1 contact residues and the functional readouts, enabling readers to directly relate chemical features to binding interactions and activity effects. To guide readers earlier in the text, we now explicitly refer to Fig. 4 at first mention of AR234958 in the Results section. The legend of Fig. 1 has been corrected accordingly. We have cross-checked all in-text figure references to ensure consistency.

Reviewer #1 – Comment 7

Fig. 2. Ligands in panels a and b should be oriented similarly, with key hydrophobic elements involved in Mas1 binding facing the same way.

Response: We thank the reviewer for this helpful suggestion. We agree that presenting the ligands in a consistent orientation will improve clarity and facilitate direct comparison. In the revised manuscript, we have corrected Figure 2 so that NPFF and AR234958 are displayed with their key hydrophobic elements aligned in the same orientation.

Reviewer #2 – Comment 1

For the various mutants it is not clear that damaging mutations did not affect cell surface expression--something easy to quantify.

Response: We thank the reviewer for raising this important point. To address this concern, we performed additional flow cytometry assays to evaluate the cell surface expression levels of all Mas1 mutants tested in this study. The results demonstrate that the mutants are expressed at levels comparable to wild-type Mas1, indicating that the observed functional changes arise from altered ligand–receptor interactions rather than reduced surface expression.

Reviewer #2 – Comment 2

The authors did not cite key refs regarding MRGPRX1 structure and function: PMID: 36302898

Response: We thank the reviewer for pointing out this missing reference. We have now cited the study (PMID: 36302898) in the revised manuscript and briefly discussed its relevance in relation to Mas1 and Gi-coupled GPCR signaling.

...MRGPRX1(Liu, Cao et al., 2023)...

Reviewer #3 – Major Concern

However, I have significant concerns regarding the quality of the density maps and the accuracy of the structural modeling-particularly for the AR234958-bound structure. Given that these structures may be used to guide drug discovery efforts, it is crucial that the models are built correctly.

The ligand density is suboptimal in both complexes, although this is a known challenge in GPCR-G protein structures, particularly in the upper portion of the receptor where flexibility often leads to poor map quality. In the case of NPFF, the overall map is reasonable, except for residues Q14 and P15. I recommend that the authors perform focused 3D classifications to explore potential conformational heterogeneity in the peptide. Considering the large particle number (>782,000), there is a good chance of identifying a subset with improved density for the NPFF-bound state. That said, even without further improvements, the current NPFF model is acceptable.

My primary concern lies with the AR234958-bound structure (PDB: sm-fitJ340-coot-18; map: P210_J340FLIP.mrc). First, there appears to be visible density for residues 96-98, yet they are not modeled. Given that this loop is near the ligand-binding pocket, accurate modeling is essential. Additionally, residues 255-259 do not seem to fit the map well. Most critically, the ligand modeling itself raises concerns. As shown in the figures, there are discontinuities and extra densities when comparing the ligand model to the map.

I acknowledge the difficulty in modeling due to the limited map quality. Again, focused 3D classification might help improve the local resolution. I attempted to build an alternative model for AR234958 using the current map, and while it is not perfect, certain aspects appear improved compared to the current model (see attached file). The authors need to provide strong evidence that their current model is the most accurate interpretation of the data.

Response:

We are deeply grateful to the reviewer for the meticulous assessment of our maps and for generously sharing an alternative ligand model for the AR234958-bound state. Your input directly enabled substantial improvements. Guided by these suggestions, we re-examined both complexes and revised the AR234958–Mas1 model as summarized below.

Actions taken for the AR234958-bound complex.

Local rebuilding: We rebuilt residues 96–98 and 255–259 against the experimental density and refined them with neighboring elements, improving model–map agreement at the pocket entrance.

Further improvement: We also attempted focused 3D classification to further improve the ligand density, but given the limited particle number and resolution, no subset with significantly enhanced ligand features could be identified.

Model adoption and refinement: We adopted the reviewer-proposed ligand model as a starting hypothesis, rebuilt the ligand and refined it against the experimental map together with the

surrounding residues.

Why we revised and used the reviewer's model (four independent reasons).

1. Energetics (MM-GBSA)

To further validate the accuracy of the ligand model, we compared the reviewer-provided model with our original one and carried out MMGBSA energy calculations in Schrödinger. The MMGBSA of the reviewer-provided is **-90.58**, which is significant lower than the former model(**-70.07**). These analyses indicated that the reviewer's model adopts a more chemically reasonable conformation with lower binding free energy. (Response Fig. 1a)

2. Consistency with functional data

The revised orientation rationalizes the mutational trends—implicating L87^{2.60}, Y248^{6.55}—and captures the 6'Fc “scaffold” role engaging I84^{2.57}, in line with the observed potency/efficacy changes (Response Fig. 1b).

3. Map Density Adaptation

The reviewer-proposed pose adapts to and conforms with the experimental cryo-EM map more faithfully (P210_J340FLIP.mrc): it shows contiguous, interpretable density (Response Fig. 1c).

4. More reasonable ligand posture

In the former model, FA1 and FA2 (within the 6'Fc region) adopted a strained π - π stacking requiring non-ideal torsions and producing local steric strain near the pocket entrance; the centroid distance/offset deviated from canonical stacking geometry, indicating a less stable interaction. The reviewer's model structurally more stable posture is consistent with the improved energetic profile and overall fit to the map. (Response Fig. 1d)

For the NPFF-bound structure

Following the reviewer's suggestion, we performed focused 3D classification in an attempt to improve the density of residues Q14 and P15. Despite extensive trials, no significant enhancement was observed, likely due to the intrinsic flexibility of these solvent-exposed residues at the extracellular interface. Importantly, the surrounding receptor residues and the remaining portion of NPFF are well resolved, providing a clear chemical environment that supports the correctness of our current model.

Summary

We have updated the AR234958–Mas1 coordinates and revised figures in the manuscript. The final model shows improved fit to the density, better energetic favorability, more reasonable ligand posture, and concordance with functional data—strengthening confidence in the structure for mechanistic interpretation and drug-design hypotheses. We again thank the reviewer for the invaluable guidance that led to these improvements.

Reviewer #3 – Other comments

I have a few additional comments regarding the signaling assays.

1. If I understand correctly, at very low ligand concentrations, the cAMP levels are primarily influenced by forskolin. In other words, the dose-response curves for each mutant should ideally start from a similar baseline. If this is the case, I suggest that the authors re-normalize the curves for better comparability.
2. It is unclear why the L87A mutant in Figure 4 shows an increase in signal. The authors should provide a plausible explanation for this observation.
3. The title of Supplementary Figure 6 is misleading. It should not be referred to as "AF3-bound MAS1," but rather as "AF3-predicted MAS1 structure."

Response:

1. We thank the reviewer for this valuable suggestion. Theoretically, forskolin directly stimulates adenylyl cyclase, resulting in cAMP accumulation that should establish a common baseline across all curves, such that each curve starts from ~100% before decreasing upon ligand addition. In our experimental workflow, we used the maximum and minimum values of the wild-type (WT) receptor as reference points for normalization. However, in practice, variations in receptor expression levels, cell conditions, and experimental handling can introduce discrepancies. As a result, some mutants displayed noticeably larger or smaller response windows than WT.

It is also possible that certain mutations alter the receptor's basal or ligand-independent signaling capacity, thereby affecting the apparent dynamic range. Nevertheless, our primary focus in this study is not on the baseline, but rather on the relative changes in activation capacity and the window size among mutants compared with the wild-type receptor. To ensure consistency, we have reanalyzed the data by normalizing all dose-response curves to the forskolin-induced baseline, which provides a uniform reference and allows clearer comparison of the ligand-induced signaling properties of each mutant.

2. Upon re-examination, we found that the apparent increase in signaling observed for the L87A mutant was due to the instability of the dose - response curve. This instability likely arose because substitution at this highly conserved position led to near-complete loss of receptor activity, making the fitted curve unreliable.

To ensure rigor, we performed additional functional assays for the L87A mutant. In these new experiments, we observed that the previously noted increase in signal was no longer present, confirming that the original result was an artifact caused by curve instability rather than a genuine gain of function. We have updated the corresponding figure and description accordingly.

3. We thank the reviewer for pointing this out. We agree that our original labeling was misleading. In the revised manuscript, we have corrected the figure title and corresponding references in the text to clearly state that this model represents an AlphaFold3-predicted MAS1 structure rather than an experimental structure.

Reference

- Ballesteros JA, Weinstein H (1995) [19] Integrated methods for the construction of three-dimensional models and computational probing of structure-function relations in G protein-coupled receptors. In *Methods in Neurosciences*, Sealfon SC (ed) pp 366-428. Academic Press
- Chikumi H, Vázquez-Prado J, Servitja JM, Miyazaki H, Gutkind JS (2002) Potent activation of RhoA by Galpha q and Gq-coupled receptors. *The Journal of biological chemistry* 277: 27130-4
- Gavard J, Gutkind JS (2008) Protein kinase C-related kinase and ROCK are required for thrombin-induced endothelial cell permeability downstream from Galpha12/13 and Galpha11/q. *The Journal of biological chemistry* 283: 29888-96
- Guo L, Zhang Y, Fang G, Tie L, Zhuang Y, Xue C, Liu Q, Zhang M, Zhu K, You C, Xu P, Yuan Q, Zhang C, Liu L, Rong N, Peng S, Liu Y, Wang C, Luo X, Lv Z et al. (2023) Ligand recognition and G protein coupling of the human itch receptor MRGPRX1. *Nature communications* 14: 5004
- He Q, Yuan Q, Shan H, Wu C, Gu Y, Wu K, Hu W, Zhang Y, He X, Xu HE, Zhao L-H (2024) Mechanisms of ligand recognition and activation of melanin-concentrating hormone receptors. *Cell Discovery* 10: 48
- Li C, Liu H, Li J, He X, Zhu H, Fu W, Xu HE (2024) Molecular basis of ligand recognition and activation of the human succinate receptor SUCR1. *Cell Research* 34: 594-596
- Liu H, Guo S, Dai A, Xu P, Li X, Huang S, He X, Wu K, Zhang X, Yang D, Xie X, Xu HE (2024) Structural and functional evidence that GPR30 is not a direct estrogen receptor. *Cell Research* 34: 530-533
- Liu Y, Cao C, Huang X-P, Gumpfer RH, Rachman MM, Shih S-L, Krumm BE, Zhang S, Shoichet BK, Fay JF, Roth BL (2023) Ligand recognition and allosteric modulation of the human MRGPRX1 receptor. *Nature chemical biology* 19: 416-422
- Sureshkumar P, Souza Dos Santos RA, Alenina N, Mergler S, Bader M (2023) Angiotensin-(1-7) mediated calcium signalling by MAS. *Peptides* 165: 171010
- Zhang M, Gui M, Wang ZF, Gorgulla C, Yu JJ, Wu H, Sun ZJ, Klenk C, Merklinger L, Morstein L, Hagn F, Plückthun A, Brown A, Nasr ML, Wagner G (2021) Cryo-EM structure of an activated GPCR-G protein complex in lipid nanodiscs. *Nature structural & molecular biology* 28: 258-267

Dr. Huaqiang Eric Xu
Shanghai Institute of Materia Medica
Chinese Academy of Sciences
Shanghai
China

6th Nov 2025

Re: EMBOJ-2025-121709R
Structural Insight into Ligand Binding and Activation of the GPCR Orphan Receptor Mas1

Dear Drs. Xu, Yang and Zhang,

Thank you for submitting your revised manuscript to The EMBO Journal. Two of the original referees have now reviewed it once more, and their comments are copied below. Both of them appreciate your revisions and improvements to the study, but referee 3 still retains important concerns with regard to the quality of the structural data and the experimental support for the models derived from it. I feel that further attempts to address these issues, by including Molecular Dynamics simulations and some analysis of additional residues, as suggested by the reviewer, would be important to strengthen the study prior to publication. I have therefore decided to return the manuscript to you for an exceptional second round of revision, which should hopefully allow you to satisfactorily respond to referee 3's remaining points.

When preparing a re-revised manuscript, I would appreciate if you could also already incorporate the following editorial issues, which would greatly facilitate processing of the manuscript in case we should eventually proceed with its publication:

- We require institutional email (rather than freemail) accounts to be listed for all (co-)corresponding authors - this information is currently missing for Dr. Yumu Zhang.
- Please double-check to make sure to all relevant funding information in the manuscript is also entered into our submission system. Currently missing in the submission system are:
the National Key R&D Program of China (2022YFC2703105, 2023YFA1800804); CAS Strategic Priority Research Program (XDB37030103, XDB1060402); Shanghai Municipal Science and Technology Major Project (2019SHZDZX02); the Lingang Laboratory (LG-GG-202204-01); State Key Laboratory of Drug Research (SKLDR-2023-TT-04)
- As we are switching from a free-text author contribution statement towards a more formal statement based on Contributor Role Taxonomy (CRediT) terms, please remove the present Author Contribution section and instead specify each author's contribution(s) directly in the Author Information page of our submission system during upload of the final manuscript. See <https://casrai.org/credit/> for more information.
- Please rename the Conflict of Interest section into "Disclosure and Competing Interests Statement", in accordance with our updated Guide to Authors (<https://www.embopress.org/competing-interests>)
- Please correct the reference list, making sure that for references with more than 10 authors, only the first up to 10 authors should be listed, followed by 'et al.' after that (please refer to our Guide to Authors for additional information on EMBO J reference format).
- When referencing individual figure panels from the text, please make sure to do so in a sequential manner (i.e., Fig 1A before 1B before 1C...). Also, please double-check that each data item is referenced at least once in the text, e.g. Fig. 5C and Appendix Tables S3 and S4 appear currently not to be called-out at all. Also, please reference all Appendix Figure sub-panels, either individually or together (e.g. "Appendix Fig S2A-D).
- On the title page of the Appendix PDF, please state "Appendix for [manuscript title], and add page numbers to the table-of-contents listing. For the Appendix figures, please place the respective figure legends always directly under each figure.
- When uploading the Figure Source Data, please make sure to combine the Source Data for Appendix Figures in a single ZIP archive; only for the main Figures one individual ZIP archive per figure should be uploaded.
- During routine pre-acceptance checks, our data editors have raised the following queries regarding figures, data, and legends; I would appreciate if you briefly answered to them in the cover letter of your final submission, and made the requested text modifications with changes/additions highlighted via the "Track changes" option, to facilitate our final checking"
 1. Please note that information related to n is missing in the legends of figures 3E, F; 4E, F
 2. Please note that the error bars are not defined in the legends of figures 3E, F; 4E, F

Finally, please provide suggestions for a short 'blurb' text prefacing and summing up the conceptual aspect of the study in two sentences (max. 250 characters), followed by 3-5 one-sentence 'bullet points' with brief factual statements of key results of the paper; they will form the basis of an editor-written 'Synopsis' accompanying the online version of the article. Please also upload a synopsis image, which can be used as a "visual title" for the synopsis section of your paper. The image should be in PNG or JPG format, and please make sure that it remains in the modest dimensions of (exactly) 550 pixels wide and 300-600 pixels high.

I am therefore returning the manuscript to you for a final round of revision, with the link below for eventual resubmission. Should you have any questions regarding the referee comments or this decision, please do not hesitate to contact me directly.

With kind regards,

Hartmut

- 1) Every manuscript requires a Data Availability section (even if only stating that no deposited datasets are included). Primary datasets or computer code produced in the current study have to be deposited in appropriate public repositories prior to resubmission, and reviewer access details provided in case that public access is not yet allowed. Further information: embopress.org/page/journal/14602075/authorguide#dataavailability
- 2) Each figure legend must specify
 - size of the scale bars that are mandatory for all micrograph panels
 - the statistical test used to generate error bars and P-values
 - the type error bars (e.g., S.E.M., S.D.)
 - the number (n) and nature (biological or technical replicate) of independent experiments underlying each data point
 - Figures may not include error bars for experiments with $n < 3$; scatter plots showing individual data points should be used instead.
- 3) Revised manuscript text (including main tables, and figure legends for main and EV figures) has to be submitted as editable text file (e.g., .docx format). We encourage highlighting of changes (e.g., via text color) for the referees' reference.
- 4) Each main and each Expanded View (EV) figure should be uploaded as individual production-quality files (preferably in .eps, .tif, .jpg formats). For suggestions on figure preparation/layout, please refer to our Figure Preparation Guidelines: <http://bit.ly/EMBOPressFigurePreparationGuideline>
- 5) Point-by-point response letters should include the original referee comments in full together with your detailed responses to them (and to specific editor requests if applicable), and also be uploaded as editable (e.g., .docx) text files.
- 6) Please complete our Author Checklist, and make sure that information entered into the checklist is also reflected in the manuscript; the checklist will be available to readers as part of the Review Process File. A download link is found at the top of our Guide to Authors: embopress.org/page/journal/14602075/authorguide
- 7) All authors listed as (co-)corresponding need to deposit, in their respective author profiles in our submission system, a unique ORCID identifier linked to their name. Please see our Guide to Authors for detailed instructions.
- 8) Please note that supplementary information at EMBO Press has been superseded by the 'Expanded View' for inclusion of additional figures, tables, movies or datasets; with up to five EV Figures being typeset and directly accessible in the HTML version of the article. For details and guidance, please refer to: embopress.org/page/journal/14602075/authorguide#expandedview
- 9) To facilitate reproducibility and cross-laboratory adoption of methodologies, please structure the Materials & Methods section as outlined in our guide to authors, including a completed Reagents and Tools Table that can be downloaded from our author guidelines as well (<https://www.embopress.org/page/journal/14602075/authorguide#structuredmethods>).

10) Digital image enhancement is acceptable practice, as long as it accurately represents the original data and conforms to community standards. If a figure has been subjected to significant electronic manipulation, this must be clearly noted in the figure legend and/or the 'Materials and Methods' section. The editors reserve the right to request original versions of figures and the original images that were used to assemble the figure. Finally, we generally encourage uploading of numerical as well as gel/blot image source data; for details see: embopress.org/page/journal/14602075/authorguide#sourcedata

Further information is available in our Guide For Authors:

In the interest of ensuring the conceptual advance provided by the work, we recommend submitting a revision within 3 months (4th Feb 2026). Please discuss the revision progress ahead of this time with the editor if you require more time to complete the revisions. Use the link below to submit your revision:

Link Not Available

Referee #1:

The authors presented two cryo-EM structures of Mas1 activated by natural peptide NPFF and synthetic molecule AR234958, both in complex with a chimeric Gi. These structures reveal the basis of Mas1 activation by disparate agonists, the only common feature of which is hydrophobicity. The mechanism of the interactions of ligands with Mas1 was confirmed by receptor mutagenesis. The structures suggest that the mechanism of Mas1 activation is distinct from many class A GPCRs. This is the major novel aspect of this study. The study is rigorous, and the results are important for the GPCR field. The manuscript was significantly improved in revision. I believe all reviewers' concerns were satisfactorily addressed.

Referee #3:

The authors have made substantial efforts to improve the map and modeling of Mas1-AR234958, and I am pleased that they have adopted my previous suggestion regarding the alternative modeling of AR234958. However, the electron density for the ligand remains suboptimal, raising concerns about the accuracy of the current model.

I fully appreciate the challenges associated with improving electron density, and it is plausible that the weak density arises from inherent flexibility of the ligand. Nevertheless, to ensure transparency, I recommend that the authors acknowledge this limitation in the main text, so readers can properly assess the reliability of the structural interpretation. Furthermore, I suggest performing molecular dynamics (MD) simulations to investigate whether the ligand exhibits dynamic behavior within the binding pocket. More interestingly, such analyses could reveal whether the observed flexibility correlates with the quality of the electron density: for instance, whether regions of the ligand with higher mobility correspond to areas with weaker density. This would provide valuable mechanistic insight and strengthen the structural interpretation.

Other issues include:

1, It is surprising that the S109A mutation has such a pronounced effect on NPFF activity (Figure 3e). If the authors propose that S109 is part of a hydrophobic pocket, mutating serine to alanine should preserve the hydrophobic character. However, the significant loss of activity upon removal of the hydroxyl group in S109A contradicts the idea that S109 solely contributes to a hydrophobic subpocket. Instead, this strongly suggests that the hydroxyl group of S109 plays a direct role in NPFF recognition. The authors should address this discrepancy and provide a mechanistic explanation for the observed functional impact.

2. The authors state that "The 3F residue of NPFF engages a deeper hydrophobic sub-pocket comprising Mas1 residues I391.39, I842.57, L872.60, Y912.64, and L2667.39." However, it is unclear why H263 and H36 are not included in this analysis, especially given that both residues are within 3.5 Å of the 3F residue, closer than I84 and L266.

3, The authors claim that "the central 2L residue bridges these sub-pockets, interacting closely with Y2486.55 and H2627.35 to stabilize NPFF's conformation within the pocket." However, structural analysis indicates that the distance between the 2L residue and H2627.35 is over 5 Å, which exceeds the typical cutoff for a close interaction. The authors should revise this statement or provide further justification for their interpretation.

4, I am unable to see how the SA group of AR234958 interacts with Y168, as the distance between these two entities in the new

model is approximately 6 Å. Additionally, regarding the interaction between the FA2 moiety and the receptor, it is unclear why residues S109, F112, and L113 were not included in the analysis.

5, I also find it difficult to follow the logic by which the authors arrive at the concept of a "ligand-induced hydrophobic compression plane" (line 259). In the preceding paragraph, they discuss mutations of H262 and H263, yet suddenly invoke "these observations" as support for the proposed hydrophobic compression plane. In my view, this hypothesis would be more appropriately introduced after the structural comparison between active Mas1 and the AlphaFold3-predicted inactive state. Either I have missed a key point, or the authors have not clearly explained the rationale behind this claim. Further clarification is needed here.

6, I am uncertain how to interpret the data for the H262A and H263A mutations. It appears that these mutations do not significantly affect the EC_{50} but instead alter the maximal response. This difference may be more reflective of variations in receptor expression levels rather than a direct effect on ligand efficacy. More importantly, as shown in Appendix Figure S5a, the baseline cAMP levels appear to differ between H262A and both the wild-type and H263A constructs. Although Table S3 and S4 report similar expression levels for H262A and H263A, it is important to note that expression levels and cAMP assays are measured using different methodologies with distinct amplification dynamics. Therefore, discrepancies between functional responses and expression data should be interpreted with caution.

7, In Figure 2c, the orientation appears to show a $\sim 180^\circ$ rotation rather than the labeled 90° rotation. Additionally, it would be much clearer for readers if the panels were labeled as "intracellular view" and "extracellular view".

8, The arrow in Figure 5a appears to be pointing in the wrong direction. Please double-check.

Response to Reviewers – EMBO Journal Submission**Meta Information****Manuscript Title: Structural Insight into Ligand Binding and Activation of the GPCR Orphan Receptor Mas1****Manuscript ID: Manuscript EMBOJ-2025-121709****Corresponding Author: Yumu Zhang, Dehua Yang, H.Eric Xu****Revision Round: Round 2****Submission Date: 20260304**

In this revised version, we have carefully addressed all reviewer comments:

© Conducted comprehensive model refinement and validation using molecular dynamics (MD) simulations to corroborate the structural stability of the ligand-receptor complex and rationalized local electron density features.

© Re-evaluated the precise functional roles of key transmembrane residues in ligand recognition versus receptor activation, resolving discrepancies between structural observations and mutagenesis data.

© Strengthened the interpretation of signaling efficacy by providing rigorous control data on receptor surface expression, ensuring that functional impairments are attributed to intrinsic mechanistic defects rather than experimental variations.

© Restructured the manuscript sections to articulate the activation mechanism more effectively, providing a coherent logical progression from structural evidence to our proposed conceptual hypothesis.

© Enhanced the clarity and accuracy of visual data presentation, corrected figure orientations, and strictly adhered to all editorial and formatting guidelines to ensure the highest standard of reporting.

We believe that these revisions substantially improve the rigor, accuracy, and clarity of our work. We are grateful to the reviewers for their insightful guidance, which has helped us strengthen the manuscript.

Reviewer #1

Comment: *...The authors presented two cryo-EM structures of Mas1 activated by natural peptide NPFF and synthetic molecule AR234958, both in complex with a chimeric Gi. These structures reveal the basis of Mas1 activation by disparate agonists, the only common feature of which is hydrophobicity. The mechanism of the interactions of ligands with Mas1 was confirmed by receptor mutagenesis. The structures suggest that the mechanism of Mas1 activation is distinct from many class A GPCRs. This is the major novel aspect of this study. The study is rigorous, and the results are important for the GPCR field. The manuscript was significantly improved in revision. I believe all reviewers' concerns were satisfactorily addressed.*

Response: We thank the reviewer for their positive assessment of our work and for highlighting the significance of our findings regarding the distinct hydrophobic activation mechanism of Mas1. We are pleased that the revisions have satisfactorily addressed the reviewer's previous concerns.

Reviewer #3

Major Comment: *...The authors have made substantial efforts to improve the map and modeling of Mas1-AR234958, and I am pleased that they have adopted my previous suggestion regarding the alternative modeling of AR234958. However, the electron density for the ligand remains suboptimal, raising concerns about the accuracy of the current model.*

I fully appreciate the challenges associated with improving electron density, and it is plausible that the weak density arises from inherent flexibility of the ligand. Nevertheless, to ensure transparency, I recommend that the authors acknowledge this limitation in the main text, so readers can properly assess the reliability of the structural interpretation. Furthermore, I suggest performing molecular dynamics (MD) simulations to investigate whether the ligand exhibits dynamic behavior within the binding pocket. More interestingly, such analyses could reveal whether the observed flexibility correlates with the quality of the electron density: for instance, whether regions of the ligand with higher mobility correspond to areas with weaker density. This would provide valuable mechanistic insight and strengthen the structural interpretation.

Response: We sincerely thank the reviewer for this insightful comment. It has been pivotal in refining our structural model.

1. Model Refinement via MD Simulations: Initially, preliminary MD simulations of our previous model suggested significant instability, indicating potential inaccuracies in the ligand placement. Guided by these simulations and our functional data, we refined the binding pose of AR234958. Based on this new model, we performed triplicate independent 200-ns MD simulations.
2. Stability: The refined model demonstrated robust stability across all three replicas, maintaining a consistent binding pose with RMSD values plateauing around 4–6 Å (Appendix Figure S5a). This confirms that our new model is energetically favorable and reliable.
3. Correlation with Density: Furthermore, the per-atom Root Mean Square Fluctuation (RMSF) analysis (Appendix Figure S5c) revealed significant conformational flexibility in the peripheral aromatic moieties. Crucially, mapping these high-mobility regions onto the cryo-EM map shows a

striking correlation with the areas of weaker electron density (Appendix Figure S5a).

4. Acknowledgment of Limitation: As recommended, we have included a specific statement in the main text explicitly acknowledging the suboptimal electron density for these flexible regions. This ensures transparency while providing a physical explanation for the local map quality.

Changes in Manuscript:

Structural Model: We have updated the structural model of the Mas1–AR234958 complex throughout the manuscript and figures (specifically Figure 4 and Figure 5) to reflect the refined binding pose validated by MD.

Text: Added a paragraph describing the model refinement via MD and acknowledging the limitation of local conformational flexibility. The sentence showed as below:

To further validate the binding stability and rationalize the local electron density quality, we performed triplicate independent 200-ns molecular dynamics (MD) simulations of the Mas1–AR234958 complex. The simulations demonstrated that the ligand maintains a stable binding pose within the orthosteric pocket (Appendix Figure S5a). Notably, structural mapping of the ligand flexibility onto the cryo-EM density map (Appendix Figure S5b) revealed that the regions of high mobility correspond precisely to the areas of weaker electron density. This observation is further corroborated by per-atom Root Mean Square Fluctuation (RMSF) analysis, which identified significant conformational flexibility in the peripheral aromatic moieties (Appendix Figure S5c). Given this dynamic behavior, the structural model in these flexible regions should be interpreted with the acknowledged limitation of local conformational heterogeneity.

Figures: Added Appendix Figure S5, presenting the stable MD trajectories (RMSD), structural mapping of flexibility vs. density, and the RMSF profile of the refined model

Other Comments

Comment 1: *...It is surprising that the S109A mutation has such a pronounced effect on NPPF activity (Figure 3e). If the authors propose that S109 is part of a hydrophobic pocket, mutating serine to alanine should preserve the hydrophobic character. However, the significant loss of activity upon removal of the hydroxyl group in S109A contradicts the idea that S109 solely contributes to a hydrophobic subpocket. Instead, this strongly suggests that the hydroxyl group of S109 plays a direct role in NPPF recognition. The authors should address this discrepancy and provide a mechanistic explanation for the observed functional impact.*

Response: We sincerely thank the reviewer for this insightful comment. We agree that the significant loss of activity in the S109A mutant contradicts the notion that S109 contributes solely to a hydrophobic environment.

1. Structural Re-analysis: Upon re-examining the interface, we found that the hydroxyl group of S109 forms a hydrogen bond (3.55 Å) with the backbone amide nitrogen of the first residue 1F of

NPFF. This interaction acts as a critical "polar anchor" for the peptide N-terminus, which is essential for proper ligand positioning.

2. Mechanistic Explanation: The S109A mutation removes this hydroxyl group, disrupting the hydrogen bond and destabilizing peptide binding. This explains the pronounced reduction in potency observed in our functional assays (Figure 3e).

Changes in Manuscript:

We have revised the description to highlight the polar interaction of S109, rather than grouping it solely with hydrophobic residues.

Figure: We have updated Figure 3b to explicitly visualize this specific hydrogen bond interaction between the side chain of S109 and the backbone nitrogen of NPFF-1F.

Comment 2: *...The authors state that "The 3F residue of NPFF engages a deeper hydrophobic sub-pocket comprising Mas1 residues I39^{1.39}, I84^{2.57}, L872.60, Y91^{2.64}, and L266^{7.39}." However, it is unclear why H263 and H36 are not included in this analysis, especially given that both residues are within 3.5 Å of the 3F residue, closer than I84 and L266.*

Response: We sincerely thank the reviewer for this detailed observation. We confirm that residues H36^{1.36} and H263^{7.36} are spatially proximal to the 3F residue of NPFF in our model.

Correction & Functional Context:

Structural Update: We have added both residues to the structural description for completeness.

Functional Distinction: However, our mutagenesis data reveal a clear functional dichotomy:

H36^{1.36}: The H36A mutation reduced maximal response (E_{max}), confirming its functional importance (likely in pocket maintenance).

H263^{7.36}: In contrast, the H263A mutation had no significant effect on ligand potency or efficacy (Appendix Figure S6, Appendix Table S2). This indicates that despite its proximity, H263 does not contribute significantly to ligand recognition or activation, cautioning against overinterpreting its structural position.

Changes in Manuscript:

We have updated the text to include these residues but explicitly noted that H263 appears functionally dispensable based on mutagenesis data. The sentence shows as below:

The H36A mutation primarily reduced maximal response (E_{max}) without significantly shifting ligand potency (Fig. 3g, Appendix Table S2). Interestingly, the H263A mutation had minimal impact on receptor activity (Appendix Table S3). This suggests that while H36^{1.36} structurally lines the pocket, its role—analogue to H262^{7.35}—is likely centered on maintaining the active pocket conformation rather than determining ligand binding affinity. Furthermore, although H263^{7.36} is spatially proximal, it appears functionally dispensable for NPFF recognition.

Comment 3: *...The authors claim that "the central 2L residue bridges these sub-pockets, interacting closely with Y248^{6.55} and H262^{7.35} to stabilize NPFF's conformation within the pocket." However, structural analysis indicates that the distance between the 2L residue and*

H262^{7.35} is over 5 Å, which exceeds the typical cutoff for a close interaction. The authors should revise this statement or provide further justification for their interpretation.

Response: We thank the reviewer for this precise structural check. We have re-examined the interface in our refined model and agree that H262^{7.35} does not form a close direct contact with the 2L residue of NPFF.

Correction & Insight: We have revised the text to accurately reflect this observation. Interestingly, this structural finding aligns remarkably well with our functional mutagenesis data (see response to Comment 9).

Structure: H262^{7.35} defines the lower boundary of the binding pocket but lacks direct high-affinity interactions with the ligand.

Function: The H262A mutation drastically reduces maximal activation (E_{max}) without significantly altering ligand potency (EC₅₀) (Appendix Table S2).

Synthesis: This suggests that H262 is not primarily responsible for ligand binding affinity (consistent with the lack of direct contact and stable EC₅₀), but is crucial for maintaining the structural integrity of the active pocket or facilitating the conformational transition required for activation (hence the profound loss of E_{max}).

Changes in Manuscript:

We have modified the description to state that H262 defines the structural boundary of the sub-pocket to support NPFF and AR234958' conformations, removing the claim of a "close interaction."

NPFF section: We changed: *The central 2L residue bridges these sub-pockets, interacting closely with Y248^{6.55}, while H262^{7.35} defines the structural boundary of the sub-pocket to support NPFF's conformation (Fig. 3c). Mutagenesis of Y248^{6.55} and H262^{7.35} markedly reduced ligand efficacy. Notably, although flow cytometry confirmed robust surface expression of the H262A mutant (~90% of WT, Appendix Table S2), it caused a drastic reduction in maximal response (E_{max}) without significantly altering ligand potency (EC₅₀). This phenotype suggests that H262 is critical for maintaining the active pocket conformation or transduction efficiency, rather than mediating direct high-affinity ligand contacts.*

Comment 4: *...I am unable to see how the SA group of AR234958 interacts with Y168, as the distance between these two entities in the new model is approximately 6 Å. Additionally, regarding the interaction between the FA2 moiety and the receptor, it is unclear why residues S109, F112, and L113 were not included in the analysis.*

Response: We thank the reviewer for the detailed inspection of the model and for raising these important points regarding ligand-receptor contacts.

1. Regarding Y168^{4.61} and the SA1 moiety: We agree with the reviewer that in our refined model, the distance between the SA1 group and Y168^{4.61} is quite far, precluding a strong direct interaction. However, our mutagenesis data (Figure 4e) show that the Y168A mutation completely abolishes receptor activity. This indicates that Y168^{4.61} acts as a critical "structural gatekeeper." Its role is to maintain the structural integrity of the extracellular vestibule, ensuring the pocket is correctly shaped for ligand access and stabilization, even without direct high-affinity contact.

2. Regarding TM3 residues (S109, F112, L113) and the FA2 moiety: We appreciate the reviewer's suggestion to analyze TM3. It is important to note that based on our refined structural model, the positions of TM3 residues have been updated relative to the ligand:

Structural Re-evaluation: In the improved model, residues F112 and L113 are located at a distance that suggests minimal interaction with the FA2 moiety. Therefore, they were excluded from further functional characterization.

Targeted Mutagenesis: Instead, we identified S109, V106, I105, and Y102 as the structurally proximal residues in the TM3 region within the refined model. To validate their roles, we performed alanine scanning mutagenesis on these residues.

Functional Validation and Distinct Binding Modes (Figure R1): As shown in Figure R1, the mutagenesis results revealed a striking functional dichotomy between NPFF and AR234958 binding modes:

For NPFF (Peptide): As shown in Figure 3e, the S109A mutation causes a profound loss of potency, confirming S109 as a critical "polar anchor" for peptide recognition.

For AR234958 (Small Molecule): In sharp contrast, the S109A mutation did not significantly shift the potency (EC₅₀) of AR234958, although it reduced the maximal response (E_{max}) (Figure R1). Similarly, V106A and I105A reduced E_{max} but maintained WT-like potency, while Y102A had no effect.

Conclusion: This distinct functional profile strongly supports our structural model: S109 is not the primary determinant for AR234958 binding affinity (unlike for NPFF). The reduction in E_{max} for S109A suggests that while S109 contributes to the structural stability required for full receptor activation, it does not serve as a direct anchor for the small molecule.

Changes in Manuscript:

We have updated the text to reflect the refined structural model, clarifying that Y168 serves as a boundary determinant.

We have focused the description on the conserved core residues (TM1/2/6/7) that drive AR234958 binding and highlighted the distinct role of TM3 residues in distinguishing small-molecule vs. peptide recognition.

Figure R1. Functional characterization of TM3 residues in AR234958-induced Mas1 activation. Dose-response curves of cAMP accumulation for WT Mas1 and mutants (S109A, V106A, I105A, Y102A) stimulated by AR234958. Data are presented as mean \pm SEM (n=3). Note: Unlike the profound loss of potency observed for NPFF (Fig. 3e), the S109A mutation does not significantly shift the EC₅₀ for AR234958, though it reduces maximal efficacy (E_{max}). This indicates that S109 is not a critical binding anchor for AR234958, revealing a distinct recognition mechanism compared to the NPFF bound structure.

Comment 5: ... *I also find it difficult to follow the logic by which the authors arrive at the concept of a "ligand-induced hydrophobic compression plane" (line 259). In the preceding paragraph, they discuss mutations of H262 and H263, yet suddenly invoke "these observations" as support for the proposed hydrophobic compression plane. In my view, this hypothesis would be more appropriately introduced after the structural comparison between active Mas1 and the AlphaFold3-predicted inactive state. Either I have missed a key point, or the authors have not clearly explained the rationale behind this claim. Further clarification is needed here.*

Response: We thank the reviewer for this constructive suggestion regarding the manuscript's logical flow. We agree that introducing the "hydrophobic compression plane" hypothesis immediately after the H262 mutagenesis discussion was abrupt and lacked the necessary structural context.

Reorganization: As recommended, we have restructured the text to present the "hydrophobic compression plane" concept after the detailed comparison between the active-state structures and the AlphaFold3-predicted inactive model.

New Flow: We now first describe the key structural rearrangements upon activation, including the 16.8° outward displacement of TM6 and the coordinated shifts of core residues (Fig. 6b).

Hypothesis: Following this structural evidence, we then introduce the "ligand-induced hydrophobic compression plane" as the mechanical driver for these observed conformational changes.

Changes in Manuscript:

We removed the paragraph proposing the compression plane from its original position.

We integrated this discussion into the "Hydrophobic Relay Driven Activation Mechanism" section, placing it directly after the analysis of the active-inactive state transition. This rearrangement provides a coherent logical progression from structural observation to mechanistic hypothesis.

Comment 6: ...*I am uncertain how to interpret the data for the H262A and H263A mutations. It appears that these mutations do not significantly affect the EC₅₀ but instead alter the maximal response. This difference may be more reflective of variations in receptor expression levels rather than a direct effect on ligand efficacy. More importantly, as shown in Appendix Figure S5a, the baseline cAMP levels appear to differ between H262A and both the wild-type and H263A constructs. Although Table S3 and S4 report similar expression levels for H262A and H263A, it is important to note that expression levels and cAMP assays are measured using different*

methodologies with distinct amplification dynamics. Therefore, discrepancies between functional responses and expression data should be interpreted with caution.

Response: We sincerely thank the reviewer for this insightful and rigorous comment. We fully agree that caution is warranted when interpreting Emax changes, as they can be influenced by receptor expression levels and methodological differences between assays.

Robust Expression vs. Impaired Function: While we acknowledge the potential discrepancies between assay methodologies, our quantitative data provide strong evidence that the impairment of H262A is a genuine functional defect rather than an expression artifact.

High Expression: Our flow cytometry data confirm that the H262A mutant is robustly expressed on the cell surface, reaching ~90% of wild-type levels (Appendix Table S2: $89.55\% \pm 1.31\%$).

Low Efficacy: Despite this preserved expression, the maximal response (Emax) drops dramatically to ~25% for NPPF and ~38% for AR234958.

Conclusion: This significant discrepancy between robust expression and impaired signaling strongly suggests that H262^{7.35} is critical for the intrinsic efficacy of the receptor, rather than merely affecting receptor density.

Consistent Effect Across Ligands: Crucially, this reduction in Emax was consistently observed for both chemically distinct agonists, NPPF and AR234958. This conserved phenotype reinforces the structural observation that H262^{7.35} serves as a key residue within the shared binding pocket, essential for attributing the active state regardless of the ligand type.

Baseline Variations: Regarding the baseline cAMP differences (Appendix Figure S6a), we agree that this adds complexity. Given H262^{7.35}'s critical location within the ligand-binding pocket, it is plausible that the mutation alters the receptor's basal conformational equilibrium, contributing to the observed baseline shift.

Changes in Manuscript: We have revised the text to interpret these findings with greater precision, explicitly citing the expression data to contextualize the functional results:

NPPF section: We changed: *The central 2L residue bridges these sub-pockets, interacting closely with Y248^{6.55}, while H262^{7.35} defines the structural boundary of the sub-pocket to support NPPF's conformation (Fig. 3c). Mutagenesis of Y248^{6.55} and H262^{7.35} markedly reduced ligand efficacy. Notably, although flow cytometry confirmed robust surface expression of the H262A mutant (~90% of WT, Appendix Table S3), it caused a drastic reduction in maximal response (Emax) without significantly altering ligand potency (EC50). This phenotype suggests that H262 is critical for maintaining the active pocket conformation or transduction efficiency, rather than mediating direct high-affinity ligand contacts.*

AR234958 section: We changed: *Mutations of H262^{7.35}, Y248^{6.55} or L266^{7.39} significantly reduced receptor responsiveness (Fig. 4f). Similarly, despite comparable expression levels to the wild-type (Appendix Table S3), the H262A mutant exhibited significantly impaired efficacy (Emax) for AR234958. This consistent functional impairment across chemically distinct ligands further validates H262^{7.35} as a key structural determinant for the general activation mechanism of Mas1.*

Comment 7: *...In Figure 2c, the orientation appears to show a ~180{degree sign} rotation rather*

than the labeled 90{degree sign} rotation. Additionally, it would be much clearer for readers if the panels were labeled as "intracellular view" and "extracellular view".

Response: We sincerely thank the reviewer for this precise observation. We have re-examined Figure 2c and confirm that the view depicts a horizontal 180° rotation, rather than the previously labeled 90°.

Correction: To ensure clarity and prevent misinterpretation, we have made the following improvements to Figure 2c:

Corrected Rotation: We have replaced the rotation symbol with a more intuitive icon indicating a 180° horizontal turn.

Explicit Labeling: We have added clear viewpoint labels to help readers immediately grasp the spatial orientation of the receptor.

Comment 8: *...The arrow in Figure 5a appears to be pointing in the wrong direction. Please double-check.*

Response: We thank the reviewer for this careful observation. We have double-checked Figure 5a and agree that the arrow indicating the rotational direction was inadvertently pointing in the reverse direction.

Correction: We have corrected the arrow in Figure 5a to accurately point towards the correct direction, consistent with the structural comparison described in the text.

Dr. Huaqiang Eric Xu
Shanghai Institute of Materia Medica
Chinese Academy of Sciences
Shanghai
China

10th Mar 2026

Re: EMBOJ-2025-121709R1
Structural Insight into Ligand Binding and Activation of the Orphan GPCR Mas1

Dear Dr. Xu,

Thank you for submitting your final revised manuscript for our consideration. I am pleased to inform you that we have now accepted it for publication in The EMBO Journal.

You may qualify for financial assistance for your publication charges - either via a Springer Nature fully open access agreement or an EMBO initiative. Check your eligibility: <https://link.springer.com/journal/44318/how-to-publish-with-us>

Yours sincerely,

Hartmut Vodermaier

Please note that it is The EMBO Journal policy for the transcript of the editorial process (containing referee reports and your response letters) to be published as an online supplement to each paper. If you should prefer removal of any referee-only figures included in the point-by-point response(s), e.g. because they may still be used for future publication or because they have been reproduced from published work by others, please do let us know immediately via response email.

More information is available here: <https://link.springer.com/partners/embo-press/editorial-policies#Peer%20review>